Ji *et al. Genome Biology*        (2020) 21:161

**METHOD**                                                                                           **Open Access**

# Single-cell ATAC-seq signal extraction and enhancement with SCATE

Zhicheng Ji, Weiqiang Zhou, Wenpin Hou and Hongkai Ji[*]

*Correspondence: hji@jhu.edu
Department of Biostatistics, Johns
Hopkins Bloomberg School of
Public Health, 615 North Wolfe
Street, 21205 Baltimore, MD, USA

## Abstract

Single-cell sequencing assay for transposase-accessible chromatin (scATAC-seq) is the state-of-the-art technology for analyzing genome-wide regulatory landscapes in single cells. Single-cell ATAC-seq data are sparse and noisy, and analyzing such data is challenging. Existing computational methods cannot accurately reconstruct activities of individual cis-regulatory elements (CREs) in individual cells or rare cell subpopulations. We present a new statistical framework, SCATE, that adaptively integrates information from co-activated CREs, similar cells, and publicly available regulome data to substantially increase the accuracy for estimating activities of individual CREs. We demonstrate that SCATE can be used to better reconstruct the regulatory landscape of a heterogeneous sample.

**Keywords:** Single cell, Chromatin, scATAC-seq; Bioinformatics, Statistical modeling, Machine learning, Software; Genomics, DNase-seq, Gene regulation

## Background

A cell's regulome, defined as the activities of all cis-regulatory elements (CREs) in its genome, contains crucial information for understanding how genes' transcriptional activities are regulated in normal and pathological conditions. Conventionally, regulome is measured using bulk technologies such as chromatin immunoprecipitation coupled with sequencing (ChIP-seq [1]), DNase I hypersensitive site sequencing (DNase-seq [2]), and assay for transposase-accessible chromatin followed by sequencing (ATAC-seq [3]). These technologies measure cells' average behavior in a biological sample consisting of thousands to millions of cells. They cannot analyze each individual cell. When a heterogeneous sample (e.g., a tissue sample) consisting of multiple cell types or cell states is analyzed, these bulk technologies may miss important biological signals carried by only a subset of cells.

Recent innovations in single-cell genomic technologies make it possible to map regulomes in individual cells. For example, single-cell ATAC-seq (scATAC-seq [4, 5]) and

single-cell DNase-seq (scDNase-seq [6]) are two technologies for analyzing open chromatin, a hallmark for active cis-regulatory elements, in single cells. Single-cell ChIP-seq (scChIP-seq [7]), on the other hand, allows single-cell analysis of histone modification. Technologies for simultaneously mapping open chromatin along with other -omics modalities are also under active development (e.g., scNMT-seq [8], Pi-ATAC [9], sci-CAR [10]). These single-cell technologies enable scientists to examine a heterogeneous sample with an unprecedented cellular resolution, allowing them to systematically discover and characterize unknown cell subpopulations.

Among the existing single-cell regulome mapping technologies, scATAC-seq is the most widely used one due to its relatively simple and robust protocol and its unparalleled throughput for analyzing a large number of cells. It is adopted by the Human Cell Atlas (HCA) Consortium as a major tool for characterizing regulatory landscape of human cells ([11]).

Data produced by scATAC-seq are highly sparse. For instance, a typical human scATAC-seq dataset contains $10^2$–$10^4$ cells and $10^3$–$10^5$ sequence reads per cell. However, the number of CREs in the genome far exceeds $10^5$. Thus, in a typical cell, most CREs do not have any mapped read. For CREs with reads, the number of mapped reads seldom exceeds two (Fig. 1a,b) because each locus has no more than two copies of assayable chromatin per cell in a diploid genome. Also, existing single-cell regulome mapping technologies including scATAC-seq destroy cells during the assay. Thus, they only get a snapshot of a cell at one time point. However, molecular events such as transcription factor (TF)-DNA binding and their dissociation are temporal stochastic processes. The steady-state activity of a CRE in a cell is determined by the probability that such stochastic events occur over time. Since probability is a continuous measure, the overall activity of a CRE in a cell should be a continuous signal in principle. The sparse and nearly binary scATAC-seq data collected for each CRE at one single time point therefore cannot accurately describe the CRE's continuous steady-state activity in a cell.

The discrete, sparse, and noisy data pose significant data analysis challenges. Conventional methods developed for bulk data cannot effectively analyze single-cell regulome data [12, 13]. As a result, there is a pressing need for new computational tools for single-cell regulome analysis. Recently, several single-cell regulome analysis methods have been developed. They can be grouped into three categories based on how they deal with the sparsity (Additional file 1: Table S1).

Methods in category 1, including chromVAR [12], SCRAT [13], and BROCKMAN [14], tackle sparsity by aggregating reads from multiple CREs. Instead of analyzing each CRE, they combine reads from CREs that share either a TF binding motif, a k-mer, or a co-activation pattern in DNase-seq data from the Encyclopedia of DNA Elements (ENCODE) [15, 16]. The aggregated data on motifs, k-mers, or co-activated CRE pathways are then used as features to cluster cells or characterize cell heterogeneity. To demonstrate the effect of combining CREs, Fig.1f shows chromatin accessibility in cell line GM12878 computed using non-aggregated data at each individual CRE, and Fig. 1g shows accessibility computed using SCRAT aggregated data (i.e., average normalized read count across CREs) for each co-activated CRE pathway. After aggregation, the signal in scATAC-seq became more continuous and showed higher correlation with the bulk DNase-seq-measured accessibility. One major drawback of aggregating multiple CREs

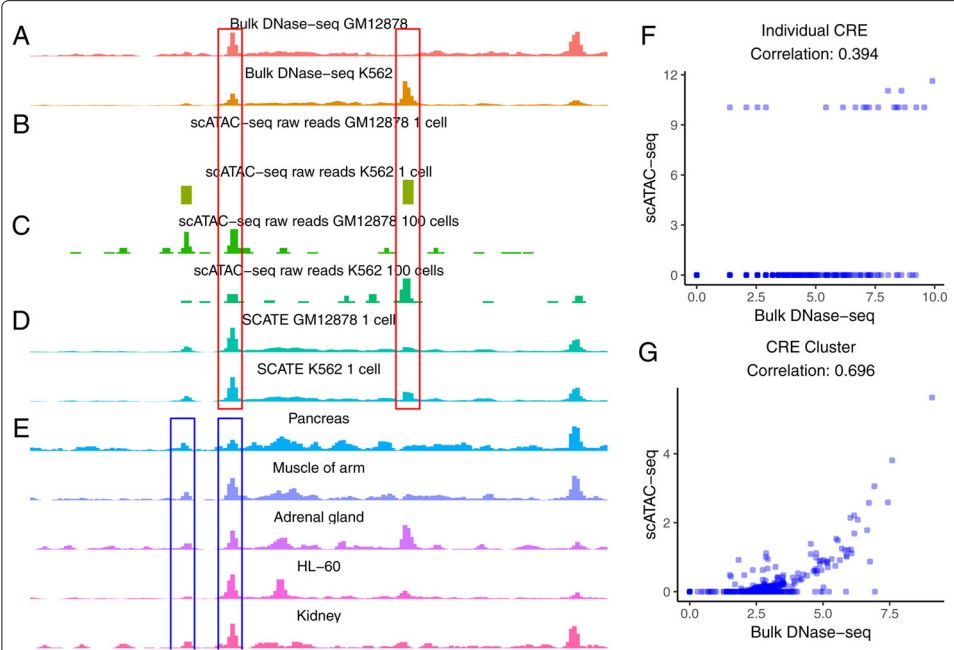

**Fig. 1** Background and motivation. **a–d** An example genomic region showing chromatin accessibility in GM12878 and K562 measured by different methods including **a** bulk DNase-seq, **b** scATAC-seq from one single cell, **c** scATAC-seq by pooling 100 cells, **d** SCATE-reconstructed scATAC-seq signal from one single cell. **e** Illustration of CRE-specific baseline activities using the same genomic region. Bulk DNase-seq data from multiple different cell types show that some loci tend to have higher activity than others regardless of cell type (e.g., compare the two loci in blue boxes). **f** At the individual CRE level, the correlation between the log-normalized scATAC-seq read count in one GM12878 cell and the log-normalized bulk GM12878 DNase-seq signal is low (Pearson correlation = 0.394). Each dot is a CRE. **g** After aggregating multiple CREs based on co-activated CRE pathways by SCRAT, the correlation between the CRE pathway activities in one GM12878 cell and the bulk GM12878 DNase-seq signal (both at log-scale) is substantially higher (Pearson correlation = 0.696). Each dot is a CRE pathway

is the loss of CRE-specific information. Thus, existing methods in this category do not analyze the activity of each individual CRE.

Methods in category 2, including Dr.seq2 [17] and Cicero [18], tackle sparsity by pooling multiple cells. Dr.seq2 [17] pools cells and applies MACS [19] to the pooled pseudobulk sample to call peaks. Cicero [18] first pools the binary chromatin accessibility profiles from similar cells to create pseudobulk samples. It then uses the pseudobulk samples to study the pairwise correlation among different CREs. Typically, scATAC-seq data pooled from multiple cells are more continuous than data from a single cell, and the pooled data also correlate better with bulk data (Fig. 1a–c). Despite this, pooling cells does not fully eliminate sparsity, particularly in a rare cell type with only a few cells. Also, pooling cells may result in loss of cell-specific information. Thus, one may want to only pool cells that are highly similar in order to better characterize a heterogeneous cell population. This could result in grouping cells into many small cell clusters, each with only a few highly similar cells. In that situation, pooling cells alone may not be enough for removing sparsity and accurately estimating activities of individual CREs.

Methods in category 3 directly work with the peak-by-cell read count matrix or its binarized version. For example, Scasat [20] converts the peak-by-cell read count matrix into a binary accessibility matrix and uses this binary matrix to cluster cells. Destin [21] applies

weighted principal components and K-means clustering to the binary accessibility matrix to cluster cells. scABC [22] uses the read count matrix to cluster cells via a weighted K-medoids clustering algorithm. PRISM [23] uses the binary accessibility matrix to compute cosine distance between cells and then uses this distance to evaluate the degree of heterogeneity of a cell population. CisTopic [24] models the binary accessibility matrix using Latent Dirichlet Allocation (LDA). This approach views each cell as a mixture of multiple topics, and each topic is a collection of peak regions and their usage preferences. The topic-cell and region-topic vectors provide a low-dimensional representation of the data. Cells and peaks are then clustered in this low-dimensional space. Category 3 methods typically are designed for specific tasks such as clustering and assessment of sample variability rather than estimating activities of individual CREs.

In summary, while existing methods provide tools for clustering cells, identifying co-accessible CREs, and analyzing sample heterogeneity, they do not address the fundamental issue of accurately reconstructing activities of each individual CRE using sparse data. Knowing activities of each individual CRE is crucial for functional studies. For example, such knowledge can be used to inform the selection of CREs for knock-out or transgenic experiments. In order to facilitate accurate reconstruction of CRE activities using scATAC-seq data, this article introduces a new statistical and analytical framework SCATE (Single-Cell ATAC-seq Signal Extraction and Enhancement). SCATE employs a model-based approach to integrate three types of information: (1) co-activated CREs, (2) similar cells, and (3) publicly available bulk regulome data. Unlike the existing methods that either aggregate CREs (category 1) or cells (category 2) but not both, SCATE combines both types of information. SCATE also uniquely uses public regulome data to enhance the analysis and adaptively optimizes the analysis resolution based on the available information in the scATAC-seq data. SCATE is freely available as an open source R package via GitHub. Compared to the existing methods, SCATE can more accurately predict CRE activities and transcription factor binding sites using the sparse data from a single cell (Fig. 1b, d) or a rare cell type as we shall demonstrate.

## Results

### SCATE model for a single cell

SCATE begins with compiling a list of candidate CREs and grouping co-activated CREs into clusters. Currently, most scATAC-seq data are generated from human and mouse. For user's convenience, for these two species we have constructed a Bulk DNase-seq Database (BDDB) consisting of normalized DNase-seq samples from diverse cell types generated by the ENCODE project. For each species, we compiled putative CREs using BDDB and clustered these CREs based on their co-activation patterns across BDDB samples. Users may augment these precompiled CRE lists by using SCATE-provided functions to (1) add and normalize their own bulk and pseudo-bulk (obtained by pooling single cells) DNase-seq or ATAC-seq samples to BDDB and then (2) re-detect and cluster CREs using the updated BDDB. These functions can also be used to create CRE database for other species. For human and mouse, saturation analyses show that BDDB covers most CREs one would discover in a new DNase-seq or ATAC-seq dataset. On average, a new sample only contributes <0.2% new CREs to our precompiled CRE lists (Additional file 2: Fig. S1). Thus, in order to save time and computation for CRE detection and clustering, users may directly use the precompiled CRE lists in BDDB without significant loss. In this

article, our analyses using SCATE are all carried out using these precompiled CREs as the input unless otherwise specified.

Given a list of CREs, their clustering structure, and scATAC-seq data from a single cell, SCATE will estimate the activity of each CRE. Let $y_{i,j}$ denote the observed read count for CRE $i$ ($i = 1, \ldots, I$) in cell $j$, and let $\mu_{i,j}$ denote the unobserved true activity. The goal is to infer the unobserved $\mu_{i,j}$ from the observed data $y_{i,j}$. We assume the following data generative model:

$$y_{i,j} \sim Poisson(L_j \mu_{i,j}^{sc})$$
$$\log(\mu_{i,j}^{sc}) = h_j(\log(\mu_{i,j}))$$
$$\log(\mu_{i,j}) = m_i + s_i \delta_{i,j}$$
$$\delta_j = \mathbf{X}\boldsymbol{\beta}_j$$

This model is illustrated in Fig. 2a. Below, we will explain the model and key components of the SCATE workflow in detail.

(1) Modeling a CRE's cell-independent but CRE-specific baseline behavior using publicly available bulk regulome data. By analyzing large amounts of ENCODE DNase-seq data, we found that these bulk data contain invaluable information not captured by the sparse single-cell data. In particular, our recent analysis of DNase-seq data from diverse cell types shows that different CREs have different baseline activities [25]. Some CREs tend to have higher activity levels than others regardless of cell type (Fig. 1e: compare two CREs in blue boxes). As a result, the mean DNase-seq profile across diverse cell types can explain a substantial proportion of data variation in the regulome profile of each individual cell type. In fact, 55.7% of the total data variance in BDDB human DNase-seq samples is explained by the mean human DNase-seq profile, and 60.1% of the total data variance

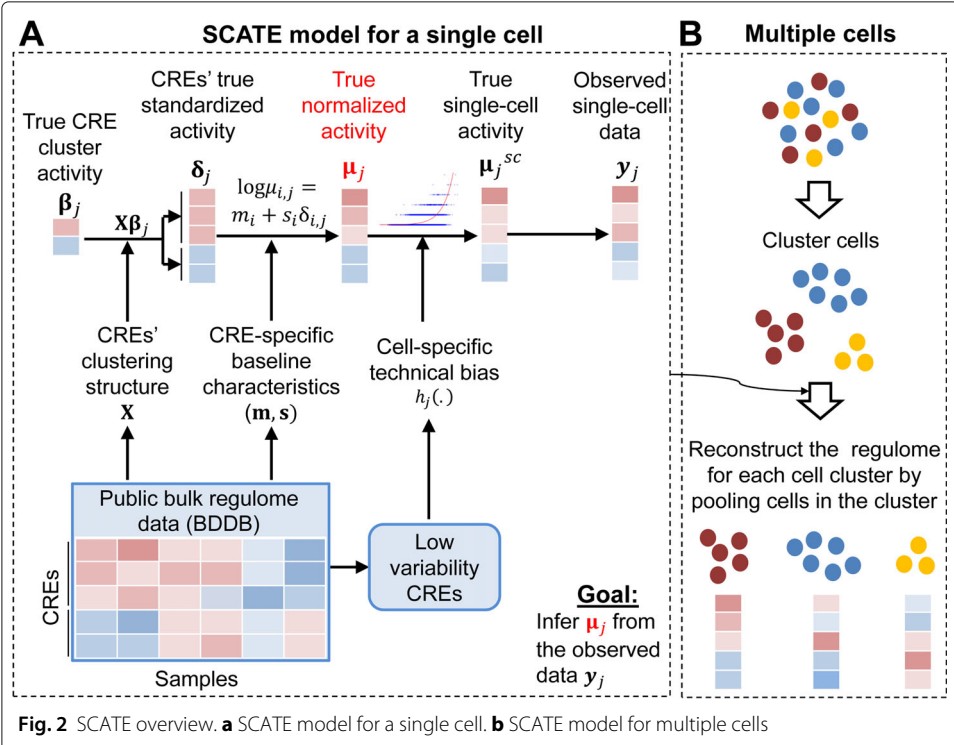

**Fig. 2** SCATE overview. **a** SCATE model for a single cell. **b** SCATE model for multiple cells

in BDDB mouse DNase-seq samples is explained by the mean mouse DNase-seq profile (Methods). The Pearson correlation coefficient between the mean DNase-seq profile and each individual DNase-seq sample in the BDDB is bigger than 0.5 for most of the samples, and the median correlation is 0.78 for human and 0.81 for mouse (Additional file 2: Fig. S2). In other words, the mean DNase-seq profile to a large extent predicts the DNase-seq profile in each individual cell type, even though the mean DNase-seq profile cannot capture cell-type-specific CRE activities. In [25], we found that the mean DNase-seq profile correlates well with independently measured TF binding activities, indicating that differences in the baseline activity among different CREs captured by the mean DNase-seq profile are real biological signals rather than technical artifacts. These highly reproducible CRE-specific baseline activities cannot be captured by the sparse data in a single cell or by pooling a small number of cells (Fig.. 1b, c, and e). Thus, in order to better reconstruct activities of each individual CRE from scATAC-seq, SCATE explicitly models these cell-type-invariant but CRE-specific baseline behaviors by fitting a statistical model to the large compendium of bulk DNase-seq data in BDDB. This allows us to estimate the baseline mean activity ($m_i$) and variability ($s_i$) of each CRE $i$.

(2) Modeling a CRE's cell-dependent activity by borrowing information from similar CREs. We model the activity of CRE $i$ in cell $j$, denoted by $\mu_{i,j}$, by decomposing it into two components: a cell-type invariant component that models the baseline behavior ($m_i$ and $s_i$), and a cell-dependent component $\delta_{i,j}$ for modeling the CRE's cell-specific activity. In other words, $\log(\mu_{i,j}) = m_i + s_i \delta_{i,j}$. The cell-type invariant component is learned from BDDB as described above. The cell-dependent component is learned using scATAC-seq data in each cell. To do so, we leverage CREs' clustering structure. Recall that co-activated CREs are grouped into clusters. We assume that CREs in the same cluster have the same $\delta_{i,j}$. Thus, information is shared across multiple co-activated CREs. Mathematically, this amounts to assuming $\boldsymbol{\delta}_j = \mathbf{X}\boldsymbol{\beta}_j$ where $\boldsymbol{\delta}_j$ is the vector of all CREs' activities in cell $j$, $\mathbf{X}$ is a binary matrix that encodes CREs' cluster membership, and $\boldsymbol{\beta}_j$ is a vector containing the activities of CRE clusters (see Methods). Unlike other methods, we only share information through $\delta_{i,j}$ rather than assuming that $\mu_{i,j}$ is the same across similar CREs. In our approach, two CREs in the same cluster have the same $\delta$, but they can have different activities (i.e., different $\mu$s) because of the difference in their CRE-specific baseline behaviors.

(3) Bulk and single-cell data normalization. Since CREs' baseline characteristics are learned from bulk DNase-seq data but our goal is to model scATAC-seq data, we need to reconcile differences between these two technologies. To do so, we assume that $\mu_{i,j}$ is the unobserved true activity of CRE $i$ in cell $j$ one would obtain if one could measure a bulk DNase-seq sample consisting of cells identical to cell $j$. In scATAC-seq data, $\mu_{i,j}$ is distorted to become $\mu_{i,j}^{sc}$ due to technical biases in scATAC-seq compared to bulk DNase-seq. These unknown technical biases are modeled using a cell-specific monotone function $h_j(.)$ such that $\log(\mu_{i,j}^{sc}) = h_j(\log(\mu_{i,j}))$. The observed scATAC-seq read count data are then modeled using Poisson distributions with mean $L_j \mu_{i,j}^{sc}$ where $L_j$ is cell $j$'s library size. The technical bias function $h_j(.)$ normalizes scATAC-seq and bulk DNase-seq data. We developed a method to estimate this unknown function by using CREs whose activities are nearly constant across diverse cell types in BDDB. Once $h_j(.)$ is estimated, CRE activities $\delta_{i,j}$ and $\mu_{i,j}$ can be inferred by fitting the SCATE model to the observed read count data.

(4) Adaptively optimizing the analysis resolution based on available data. In order to examine the activity of each individual CRE, ideally one would hope to pool as few CREs as possible. However, when data are sparse, pooling too few CREs will lack the power to robustly distinguish biological signals from noise. Thus, the optimal analysis should carefully balance these two competing needs. All existing methods reviewed in category 1 pool CREs based on fixed and predefined pathways (e.g., all motif sites of a TF binding motif). They do not adaptively tune the analysis resolution based on the amount of available information. In SCATE, co-activated CREs are grouped into $K$ clusters. Information is shared among CREs in the same cluster. We uniquely treat $K$ as a tuning parameter and developed a cross-validation procedure to adaptively choose the optimal $K$ based on the available data. When the data is highly sparse, SCATE will choose a small $K$ so that each cluster contains a large number of CREs. As a result, the activity of a CRE will be estimated by borrowing information from many other CREs. This sacrifices some CRE-specific information in exchange for higher estimation precision (i.e., lower estimation variance). When the data is less sparse and more CREs have non-zero read counts, SCATE will choose a large $K$ so that each cluster will contain a small number of CREs. As a result, the CRE activity estimation will borrow information from only a few most similar CREs, and more CRE-specific information will be retained.

(5) Postprocessing. After estimating CRE activities, we will further process all genomic regions outside the input CRE list. SCATE will transform read counts at these remaining regions to bring them to a scale normalized with the reconstructed CRE activities. The transformed data can then be used for downstream analyses such as peak calling, TF binding site prediction, or other whole-genome analyses.

### SCATE for a cell population consisting of multiple cells

For a homogeneous cell population with multiple cells, we will pool reads from all cells together to create a pseudo-cell. We will then treat the pseudo-cell as a single cell and apply SCATE to reconstruct CRE activities. Similar to Dr.seq2, this approach combines similar cells to estimate CRE activities. Unlike Dr.seq2, we also combine information from co-activated CREs and public bulk regulome data as described above. Moreover, SCATE adaptively tunes the resolution for combining CREs (i.e., the CRE cluster number $K$) which is lacking in other methods. As the cell number in the population increases, the sparsity of the pseudo-cell will decrease and the optimal analysis resolution chosen by SCATE typically will increase.

For a heterogeneous cell population, we first group similar cells into clusters. SCATE is then applied to each cell cluster to reconstruct CRE activities by treating the cluster as a homogeneous cell population (Fig.. 2b). By default, SCATE uses model-based clustering [26] to cluster cells, and the cluster number is automatically chosen by the Bayesian Information Criterion (BIC). Since one clustering method is unlikely to be optimal for all applications, we also provide users with the option to adjust the cluster number or provide their own cell clustering. SCATE can be run using user-specified cluster number or clustering results. For example, if users believe that the default clustering does not sufficiently capture the heterogeneity, they could increase the cluster number. In the most extreme case, if one sets the cluster number equal to the cell number, each cluster will become a single cell.

We note that pooling cells in each cluster to create a pseudobulk sample does not mean that the value of single-cell analysis is lost or that scATAC-seq can be replaced by bulk ATAC-seq or DNase-seq. This is because bulk ATAC-seq or DNase-seq analysis of a heterogeneous sample cannot separate different cell subpopulations or discover new cell types. Even if one could use cell sorting to separate cells in a sample by cell type and then apply bulk analysis to each cell type, the sorting relies on known cell type markers and therefore cannot discover new cell types. By contrast, a scATAC-seq experiment coupled with SCATE can identify and characterize different cell populations including potentially new cell types in a heterogeneous sample.

### Benchmark data

We compiled three benchmark datasets for method comparison. Dataset 1 consists of human scATAC-seq data from two different cell lines GM12878 (220 cells) and K562 (157 cells) generated by [4]. For this dataset, ENCODE bulk DNase-seq data for GM12878 and K562 were used as the gold standard to evaluate signal reconstruction accuracy. Dataset 2 contains scATAC-seq data from human common myeloid progenitor (CMP) cells (637 cells) and monocytes (83 cells) obtained from [27, 28]. We also obtained bulk ATAC-seq data from human CMP and monocytes generated by [28] and used them as gold standard. Dataset 3 consists of mouse scATAC-seq data from brain (3321 cells) and thymus (7775 cells) generated by [29]. For evaluation, the ENCODE bulk DNase-seq data for mouse brain and thymus were used as gold standard. In all evaluations, we removed the test cell types from the BDDB before running SCATE in order to avoid using the same bulk regulome data in both SCATE model fitting and performance evaluation.

For method evaluation, ideally one would like to have a gold standard for each single cell. However, single-cell resolution gold standard is difficult to obtain. For this reason, our method evaluation primarily relied on comparing scATAC-seq signals reconstructed from a single cell or by pooling multiple cells to bulk DNase-seq or ATAC-seq signals. In this regard, one may view single cells as random samples from a cell population, and the bulk signal characterizes cells' mean behavior in the cell population. Although each cell is not exactly the same as the population mean, its behavior should fluctuate around the mean. Moreover, one should expect that the pseudobulk signal obtained by pooling an increasing number of cells should become increasingly more similar to the true bulk signal.

### Analysis of a homogeneous cell population - a demonstration

We first demonstrate SCATE analysis of a homogeneous cell population using the GM12878 and K562 data (Dataset 1) as an example. It should be pointed out that "homogeneous" is a relative concept rather than an absolute one since one can always define cell subtypes in a cell population by computationally grouping cells into clusters and subclusters at different granularity levels. In this study, "homogeneous" is technically defined as the finest granularity level for which we were able to obtain the corresponding bulk gold standard regulome data for method evaluation. We use this technical definition for two reasons. First, even if a test cell type may potentially be decomposed further into multiple cell subtypes, we could not conduct the benchmark analysis at the cell subtype level if the gold standard bulk regulome data for those cell subtypes are unavailable and the true subtype label of each cell is unknown. Second, the primary goal of our analysis of

a homogeneous cell population is to serve as a bridge to help readers understand how SCATE would analyze each cell cluster in a heterogeneous cell population. Our working definition of "homogeneous" is sufficient to meet this need.

In this section, we applied SCATE to GM12878 and K562 separately. For each cell type, we randomly sampled $n$ ($n = 1, 5, 10, 25, 50, 100$, etc.) cells and pooled their sequence reads together to run SCATE. CRE activities reconstructed by SCATE were compared with their activities measured by bulk DNase-seq in the corresponding cell type.

Figure 3 shows the normalization function $h_j(.)$ learned by SCATE for normalizing scATAC-seq and the BDDB bulk DNase-seq data. Here, each scatter plot corresponds to a pooled scATAC-seq sample. Different plots represent different cell numbers or cell types. In these plots, each data point is a low-variability CRE with nearly constant activity across BDDB samples. For each CRE, the read count in the pooled scATAC-seq sample or the bulk ATAC-seq sample ($Y$-axis) versus the CRE's baseline mean activity in BDDB DNase-seq data ($X$-axis) is shown. The red curve is the SCATE-fitted function ($e^{h_j(.)}$) for modeling technical biases in scATAC-seq. Overall, the scATAC-seq read counts were positively correlated with CREs' baseline activities at these low-variability CREs, and the SCATE-fitted normalization functions were able to capture the systematic relationship (i.e., technical biases) between the scATAC-seq and bulk DNase-seq data. Besides scATAC-seq, we also tested SCATE's normalization algorithm in bulk data. Additional file 2: Figure S3 shows the SCATE-fitted function ($e^{h_j(.)}$) for normalizing bulk ATAC-seq data in three different cell types to the BDDB bulk DNase-seq data. The normalization functions fitted by SCATE were again able to capture the systematic relationship in the

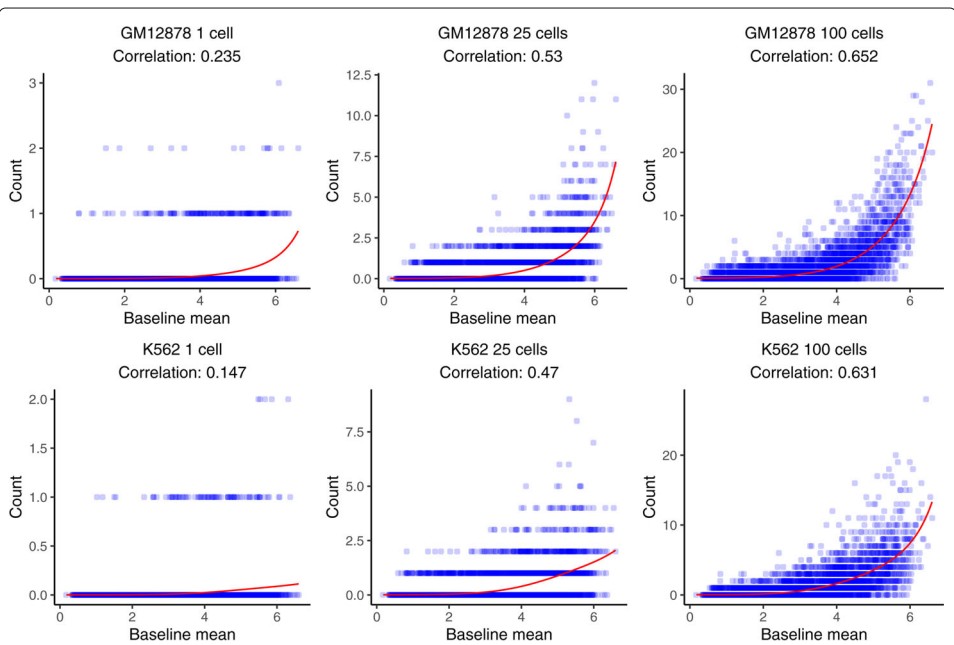

**Fig. 3** Normalization of scATAC-seq and bulk DNase-seq data. The scATAC-seq read counts versus baseline mean activities along with their Pearson correlation are shown for low-variability CREs in GM12878 (top panel) and K562 (bottom panel). Each blue dot is a low-variability CRE, defined as a CRE with almost constant activity across diverse cell types in BDDB bulk DNase-seq samples. Different plots correspond to analyses based on pooling different number of cells. In each plot, the red curve is the technical bias function fitted by SCATE

observed data, further demonstrating the effectiveness of our approach for modeling technical biases.

Figure 4 shows the number of CRE clusters adaptively chosen by SCATE. For each cell type, there are four plots corresponding to SCATE analyses by pooling different number of cells, with the cell number *n* shown on top of each plot. For each *n*, *n* cells were randomly sampled from the scATAC-seq dataset and pooled. SCATE was applied to the pooled data to automatically choose the CRE cluster number. This procedure was repeated ten times. The histogram shows the empirical distribution of the cluster number chosen by SCATE in these ten independent cell samplings without using any information from the gold standard bulk DNase-seq. As a benchmark, we also ran SCATE by manually setting the CRE cluster number *K* to different values. For each *K*, we computed the Pearson correlation between the SCATE-estimated CRE activities in scATAC-seq and the gold standard CRE activities in bulk DNase-seq. The dots in each plot show the correlation coefficients for different *K*s, also averaged across the ten independent cell samplings. The dot with the largest correlation coefficient corresponds to the true optimal cluster number. In real applications, this true optimal cluster number would be unknown because one would not have the bulk DNase-seq as the gold standard to help with choosing *K*.

Figure 4 shows that the CRE cluster number automatically chosen by SCATE (histogram) typically was close to the true optimal cluster number (the dot with the highest correlation). For instance, for analyzing a single GM12878 cell, the cluster number chosen by SCATE had its mode at 1250, and the true optimal cluster number was 2500. For analyzing 220 GM12878 cells, the cluster number chosen by SCATE had its mode at 521820, and the true optimal cluster number was also 521820.

Figure 4 also shows that, as the cell number increases, both the true optimal CRE cluster number and the cluster number chosen by SCATE also increase. Increasing the number

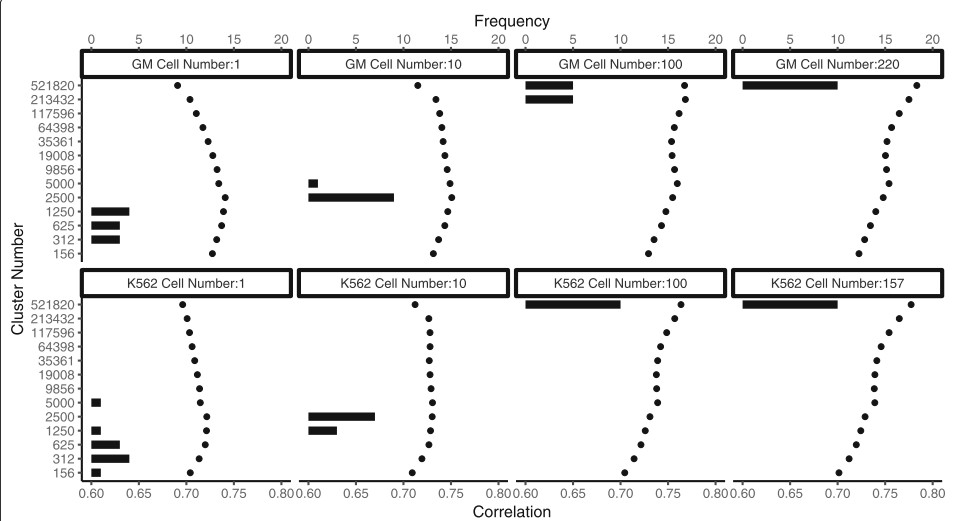

**Fig. 4** Adaptive tuning of analysis resolution. The number of CRE clusters automatically chosen by SCATE via cross-validation (histogram) is compared with the true optimal CRE cluster number determined by external information from the gold standard bulk DNase-seq data (dots). Different plots correspond to different cell types and pooled cell number. In each plot, the histogram shows the CRE cluster number chosen by SCATE in 10 independent cell samplings. The dots show the true correlation between the gold standard bulk DNase-seq signal and the SCATE-reconstructed scATAC-seq signal (both at log-scale) at each CRE cluster number, averaged across the 10 cell samplings. The dot with the highest correlation is the true optimal cluster number

of CRE clusters implies that the average number of CREs in each CRE cluster will decrease because the total number of input CREs is fixed. Thus, SCATE adaptively changes analysis resolution: as more data are available for each CRE, SCATE gradually decreases the number of CREs in each cluster for information sharing. This allows SCATE to maximally retain CRE-specific information.

Figure 5 compares SCATE-reconstructed scATAC-seq signal with bulk DNase-seq signal in GM12878 and K562 in an example genomic region. The figure has six columns corresponding to different cell types and different pooled cell numbers. For benchmark purpose, the figure also compares SCATE with a number of other methods. The data are displayed at a 200 bp non-overlapping genomic window resolution. Here "Raw reads" displays the scATAC-seq read count pooled across cells for each 200 bp genomic window.

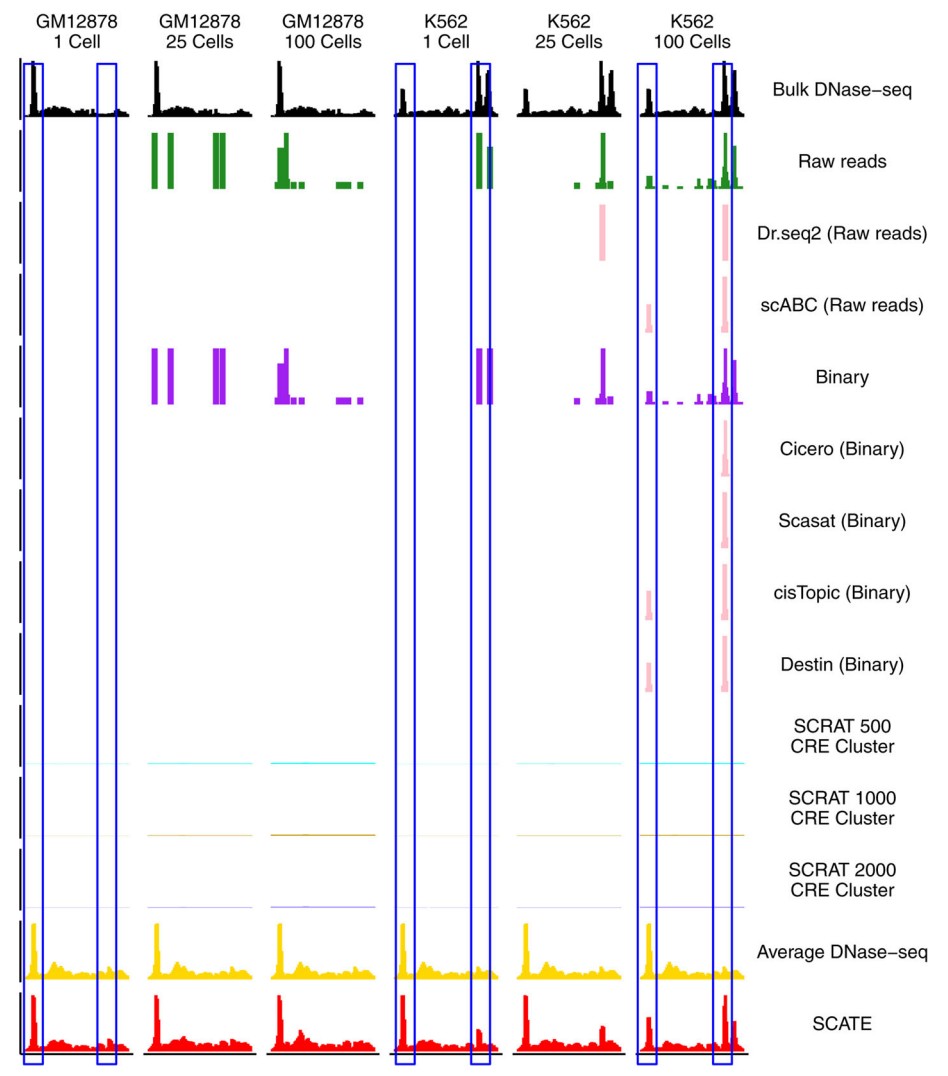

**Fig. 5** Comparison of different methods in an example genomic region. Each row is a method, each column corresponds to a different cell type or pooled cell number. All columns show the same genomic region. The blue boxes highlight two CREs. The left CRE occurs in both GM12878 and K562. It cannot be detected by Raw reads, Binary and SCRAT CRE cluster methods in a single cell, but can be detected by Average DNase-seq and SCATE. The right CRE is K562-specific. It cannot be detected by Average DNase-seq but can be detected by SCATE

"Binary" converts the 200 bp window read counts in each cell to a binary accessibility vector and then adds up the binary accessibility vectors across cells. Note that the raw read count approach is also used by Dr.seq2 and scABC to characterize CRE activities in single cells, but only in peak regions detected from the scATAC-seq data. The binary approach is also used by Cicero, Scasat, cisTopic, Destin, and PRISM to characterize CRE activities in peak regions. Since different implementations of a method may lead to variable method performance [30], we also displayed the signals obtained using these existing methods except for PRISM for which we were not able to modify its code to export the binary accessibility matrix (PRISM does not report binary accessibility as it is only used as an intermediate step to compute cell distances). Unlike these existing methods, the "Raw reads" and "Binary" methods implemented by us processed all genomic windows rather than only peak regions. ChromVAR, SCRAT, and BROCKMAN only analyze and report aggregated CRE pathway activities rather than activities of individual CREs. Thus, they cannot be compared here. However, for our previously developed SCRAT, we were able to modify the codes to estimate CRE activities by directly using pathway activities. This results in three methods, "SCRAT 500 CRE cluster," "SCRAT 1000 CRE cluster," and "SCRAT 2000 CRE cluster," shown in the figure. Here, CREs were clustered into 500, 1000, or 2000 clusters as in SCRAT using the bulk DNase-seq data in BDDB. For each CRE cluster, the average normalized scATAC-seq read count across all CREs in the cluster was calculated. It was then assigned back to each CRE in the cluster to represent the estimated CRE activity. The "Raw reads" method may be viewed as a special case of the "SCRAT CRE cluster" method when each genomic window is viewed as a CRE and each CRE forms a CRE cluster by itself (i.e., the number of CRE clusters is equal to the total number of CREs, and each cluster only contains one CRE). "Average DNase-seq" shows the average normalized read count profile of bulk DNase-seq samples in BDDB. It reflects CRE's baseline mean activity.

Figure 5 shows that SCATE-reconstructed scATAC-seq signals accurately captured the variation of CRE activities in bulk DNase-seq across different genomic loci and different cell types, whereas CRE activities estimated using raw read counts, binarized chromatin accessibility, or SCRAT CRE cluster methods all failed to accurately capture the bulk DNase-seq landscape. Interestingly, SCATE was able to use scATAC-seq data from one single cell to accurately estimate CRE activities in bulk DNase-seq. By contrast, the raw read count and binary accessibility methods both failed, likely due to data sparsity (e.g., see regions in blue boxes). The SCRAT CRE cluster method also failed, likely because (1) it assigns the same activity to all CREs in the same CRE cluster and ignores CRE-specific behaviors, and (2) it does not adaptively tune the analysis resolution as in SCATE to maximally retain CRE-specific signals. While it is also possible that signals in a single cell do not necessarily need to look like the bulk signal due to cell heterogeneity and hence explaining why signals generated by Raw reads, Binary, and CRE cluster methods in a single cell were different from the bulk signal, Fig. 5 shows that SCATE also outperformed these methods when pooling multiple cells into pseudobulk samples (e.g., pooling 25 and 100 cells), suggesting that the better performance of SCATE is real. The "Average DNase-seq" approach produced relatively continuous signals and captured some variation across genomic loci in the GM12878 and K562 bulk DNase-seq data. However, it was unable to capture cell-type-specific signals, such as those shown in the blue boxes.

### Analysis of a homogeneous cell population - a systematic evaluation

Next, we systematically evaluated SCATE and the other methods in all three benchmark datasets by treating the six test cell types as six homogeneous cell populations. The evaluation was based on the correlation with gold standard bulk regulome data, peak calling performance using reconstructed signals, and ability to predict transcription factor binding sites (TFBSs).

In the first evaluation, we computed the Pearson correlation between the scATAC-seq signals reconstructed by each method and the gold standard bulk signals across all CREs. As one example, Fig. 6a and Additional file 2: Figure S4A show the results based on pooling scATAC-seq data from 10 GM12878 cells. There are multiple methods that use the raw read counts and binary methods. For clarity of display, in this and all other analyses below, only the "Raw reads" and "Binary" methods implemented by us are shown in the main figures (e.g., Fig. 6), and the results from the other raw counts and binary methods are shown in supplementary figures (e.g., Additional file 2: Fig. S4A). Among all methods, SCATE showed the highest correlation with the bulk gold standard. We performed the same analysis on all six test cell types by pooling different cell numbers. For each cell number, we repeated the analysis ten times using ten independent cell samplings. The median performance of the ten analyses was then compared. Figure 6b and Additional file 2: Figure S4B show that SCATE consistently outperformed all the other methods and showed the strongest correlation with the bulk gold standards in all test data. When the pooled cell number was small, the improvement of SCATE over many methods was substantial. For instance, for the analysis of one single monocyte cell, the correlation was 0.01−0.22 for the different implementations of raw reads and binary methods. It was 0.57, 0.57, and 0.57 for SCRAT 500, 1000, and 2000 CRE cluster methods, respectively. For SCATE, it was 0.67, representing an improvement of 18∼6700% over the other methods. Of note, the Average DNase-seq method performed relatively well in this evaluation when the cell number was small. However, as we will show later, the average DNase-seq profile cannot predict changes in CRE activity between different cell types, but SCATE can.

In the second evaluation, we performed peak calling using scATAC-seq signals reconstucted by SCATE and other methods. Peak calling is a common task in DNase-seq or ATAC-seq data analyses. Its objective is to find genomic regions with significantly enriched signals. We implemented a peak calling algorithm using a moving average approach (see Methods) and applied it to signals reconstructed by SCATE, Raw reads, Binary, SCRAT CRE cluster, and Average DNase-seq. For the other existing raw reads and binary methods, we used their default peak calling methods to call peaks. In addition, we also performed peak calling by applying MACS2 [19] to the pseudobulk sample we obtained by pooling cells. The peak calling performance of each method was evaluated using the sensitivity versus false discovery rate (FDR) curve, where the "truth" was defined by the peaks called from the bulk gold standard data. Here, sensitivity is the proportion of true bulk peaks discovered by scATAC-seq, and FDR is the proportion of scATAC-seq peaks that are false (i.e., not found in bulk peaks). As one example, Fig. 7a and Additional file 2: Figure S5A compare the sensitivity-FDR curves of different methods when they were applied to the pooled scATAC-seq data from 25 GM12878 cells. For each curve, we computed the area under the curve (AUC). Fig. 7b and Additional file 2: Figure S5B systematically compare the AUCs of all methods in all six test cell types. In each plot, the analyses were run by pooling different numbers of cells, and the median AUC from

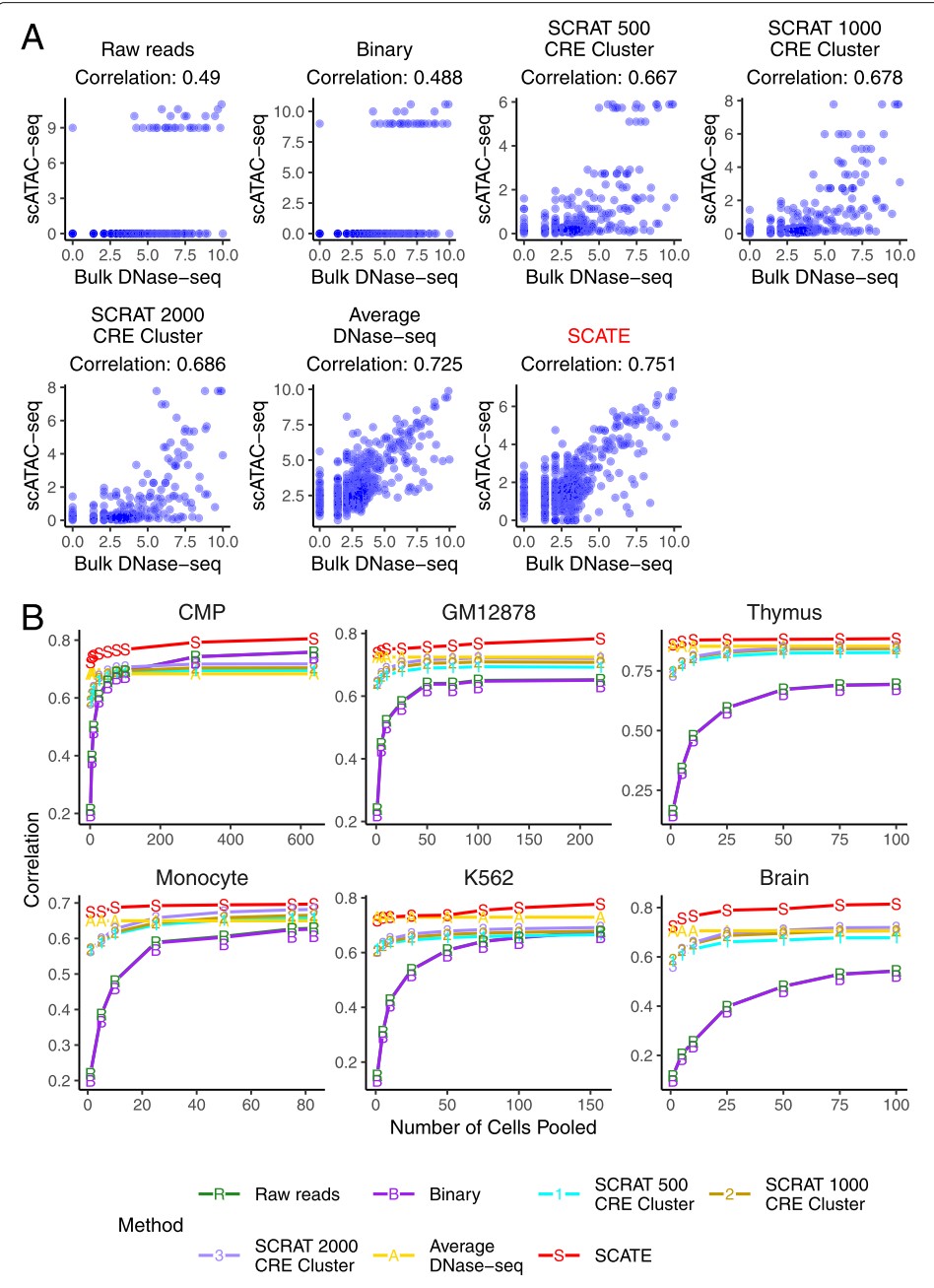

**Fig. 6** Correlation between reconstructed and true CRE activities. **a** Scatterplots showing true bulk CRE activities vs. CRE activities estimated by different methods in an analysis that pools 10 GM12878 cells. In this analysis, both activities are at log-scale **b** The correlation between the scATAC-seq reconstructed and true bulk regulome for different methods. Each plot corresponds to a test cell type. In each plot, the correlation is shown as a function of the pooled cell number

10 independent cell samplings was plotted as a function of the cell number. Once again, SCATE showed the best overall peak calling performance. When the cell number was small, the improvement was substantial. For analyzing one monocyte cell, for example, the AUC of SCATE was 0.4, whereas the AUCs for the other methods (except for Average DNase-seq) were all below 0.21. Thus, SCATE improved over these methods by 90% or more.

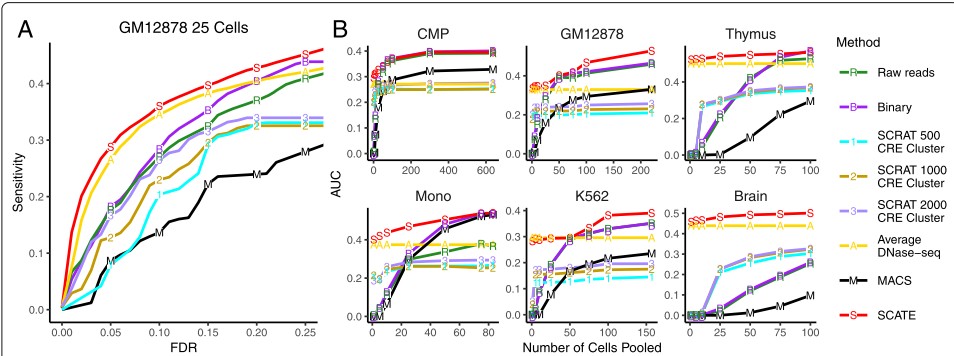

**Fig. 7** Peak calling performance. **a** The sensitivity versus FDR curve is shown for different peak calling methods in an analysis that pools 25 GM12878 cells. **b** The area under the sensitivity-FDR curve (AUC) is shown as a function of pooled cell number for different methods. Each plot corresponds to a different test cell type

In the third evaluation, we used signals reconstructed by each method to predict TFBSs. We evaluated 28 TFs in GM12878 and 29 TFs in K562 (Additional file 3: Table S2). As gold standard, we collected ChIP-seq peaks for these TFs from the ENCODE [15]. For the other cell types, we did not find TF ChIP-seq data suitable for evaluation. Therefore, our TFBS prediction analysis was focused on GM12878 and K562. To predict TFBSs of a TF, we mapped its motif sites in the genome using CisGenome [31]. Genomic windows overlapping with motif sites were sorted based on their reconstructed scATAC-seq signals. Windows with the highest signals were labeled as predicted TFBSs (Fig. 8a). Motif-containing windows that overlap with TF ChIP-seq peaks were viewed as gold standard true TFBSs. Based on this, we generated the sensitivity-FDR curve for each TF by

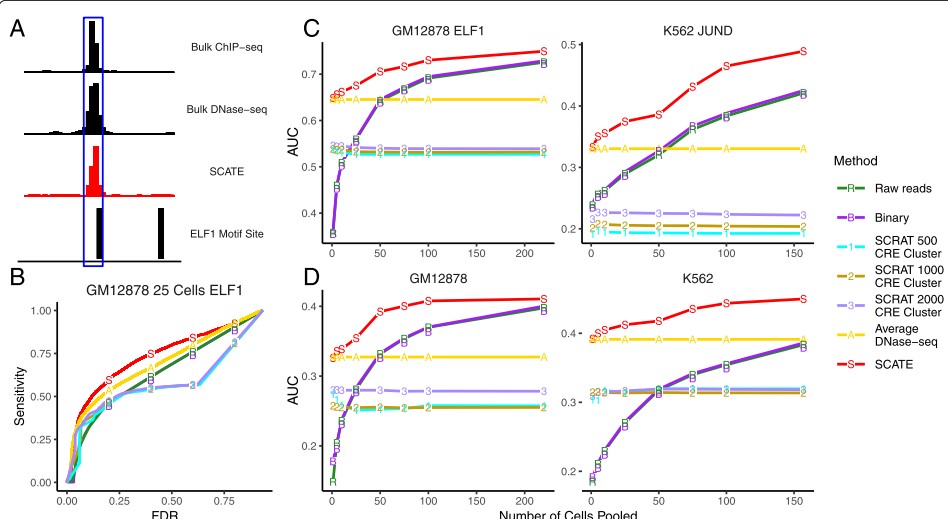

**Fig. 8** TFBS prediction performance. **a** An illustration of TFBS prediction in an example genomic region. The region contains a genomic bin with ELF1 motif and high SCATE-reconstructed CRE activity in GM12878. The bin is predicted as a ELF1 binding site. The prediction can be validated by ELF1 ChIP-seq peak in GM12878. **b** An example sensitivity versus FDR curve for comparing different methods for predicting ELF1 TFBSs in an analysis that pools 25 GM12878 cells. **c** Two examples (ELF1 in GM12878 and JUND in K562) that illustrate the method comparison across different cell numbers. In each example, analyses are performed by pooling different numbers of cells. The median AUC under the sensitivity-FDR curve from 10 independent cell samplings is shown as a function of pooled cell number. **d** The averaged AUC across all TFs is shown as a function of pooled cell number in GM12878 and K562 respectively

gradually relaxing the TFBS calling cutoff. As one example, Fig. 8b and Additional file 2: Figure S6B show the sensitivity-FDR curves of different methods for predicting ELF1 binding sites by pooling scATAC-seq data from 25 GM12878 cells. For each TF and cell type, we performed this analysis using different cell numbers. For each cell number, the median area under the sensitivity-FDR curve (AUC) of 10 independent cell samplings was computed. As two examples, Fig. 8c and Additional file 2: Figure S6C show the AUCs for different methods as a function of pooled cell number for two TFs: ELF1 in GM12878 and JUND in K562. Finally, Fig. 8d and Additional file 2: Figure S6D show the average performance of all 28 TFs in GM12878 and 29 TFs in K562. In all these analyses, SCATE robustly outperformed all the other methods. The overall improvement was substantial (e.g., see K562 in Fig. 8d and Additional file 2: Fig. S6D).

**Analysis of a heterogeneous cell population—demonstration and systematic evaluation**

The analyses of homogeneous cell populations provide a demonstration of the basic building block of SCATE. In reality, however, scATAC-seq is usually used to analyze a heterogeneous cell population consisting of multiple cell types where the cell type labels are unknown. To analyze such a heterogeneous cell population, one usually will first computationally cluster cells into relatively homogeneous subpopulations and then analyze each cell cluster as a homogeneous population. Due to inevitable noises, each cell cluster obtained in this way may not be pure. For example, while the majority of cells in a cell cluster may be of one cell type, the cluster may also contain cells from other cell types. As data analysts do not know cells' true cell type labels, they can only treat all cells in the same cluster as if they were one cell type.

  In order to see how SCATE tunes the analysis resolution when a cell cluster contains noise, we mixed K562 and GM12878 cells with different ratios (K562:GM12878 = 100%:0%, 80%:20%, 60%:40%) to mimic a cell cluster dominated by K562 cells but with different levels of noises introduced by GM12878 cells. The cross-validation procedure of SCATE was used to select the CRE cluster number as in Fig. 4. The analysis was repeated by setting the total cell number to 10 and 100 respectively. Additional file 2: Figure S7 shows that as the number of cells increased, the number of CRE clusters chosen by SCATE also increased regardless of the noise level. This indicates that when more reads are available by pooling more cells, SCATE will increase the analysis resolution. In most cases, the optimal CRE cluster numbers chosen by SCATE were largely consistent with the true optimal CRE cluster number determined by comparing scATAC-seq with bulk K562 DNase-seq. The only exception is when the noise level was high (K562:GM12878 = 60%:40% for cell number=100, where the optimal cluster number chosen by SCATE was bigger than the optimal cluster number based on bulk DNase-seq). In that case, however, one can argue that K562 bulk DNase-seq data may not reflect the chromatin profile of a mixture of K562 and GM12878 cells, whereas SCATE attempts to optimize the signal reconstruction for the cell cluster which is a mixture of K562 and GM12878 cells with almost equal proportion. Therefore, they try to measure different things and one should not expect that the optimal cluster number determined by K562 bulk DNase-seq will be consistent with the cluster number chosen by SCATE. This is different from Fig. 4 where the cell type measured by bulk gold standard is consistent with the cells analyzed by SCATE.

Next, we demonstrate how a heterogeneous cell population would be analyzed in practice. We mixed GM12878 and K562 cells from Dataset 1 with different ratios to create synthetic samples with different heterogeneity levels. Each synthetic sample had 100 cells representing a mixture of GM12878 and K562 cells. The percentage of GM12878 cells was set to $x = 10\%$, 30%, and 50%, respectively. For each percentage $x$, ten synthetic samples were created using independently sampled cells. The median performance of each method on the ten analyses was compared.

Each synthetic sample was analyzed by first clustering cells using the default cell clustering algorithm in SCATE. SCATE and other methods were then used to estimate CRE activities for each cell cluster. In all these analyses, we pretended that cells' true cell type labels were unknown and did not use them. The number of cell clusters automatically determined by SCATE in these samples ranged from 2 to 5 (Fig. 9a). Figure 9b shows one example in which cells were grouped into 2 clusters.

After running all methods, in order to evaluate whether the analysis can discover the true biology, we annotated each cell cluster using cells' true cell type labels. Each cluster was annotated based on its dominant cell type. A cell cluster was labeled as "predicted GM12878" if over 70% of cells in the cluster were indeed GM12878 cells. Similarly, a cell cluster with $\geq 70\%$ K562 cells was labeled as "predicted K562". All other clusters were labeled as "ambiguous." For a given sample, if at least one cell cluster was labeled as "predicted cell type X" (X = GM12878 or K562), we say that cell type X was detected. Based on this definition, both GM12878 and K562 can be detected in all samples (Fig. 9c). Note that one cell type may be identified by multiple cell clusters. Given the cell type annotation, we then compared the regulome of each cell type reconstructed by SCATE and other methods. Since all methods used the same cell clustering results, the comparison of their signal reconstruction ability is a fair comparison. We conducted four types of comparisons.

First, we asked whether the regulome reconstructed by each method for each predicted cell type can accurately recover the cell type's true regulome measured by the gold standard bulk data. Take GM12878 as an example. For each cell cluster predicted as GM12878, the Pearson correlation between the cluster's reconstructed scATAC-seq signal and the gold standard bulk GM12878 DNase-seq data was computed. If a sample had two or more cell clusters predicted as GM12878, each cluster was analyzed separately. The median correlation of all such clusters in ten independent synthetic samples is shown in Fig. 9d and Additional file 2: Figure S8A. SCATE again performed the best. When the proportion of GM12878 cells in a sample was small, the improvement by SCATE was larger. Figure 9e and Additional file 2: Figure S8B show the same analysis for K562, but the performance was shown as a function of GM12878 cell proportion. Figure 9f and Additional file 2: Figure S8C show the combined results. Here, at each cell mixing proportion, the median scATAC-bulk correlation of all cell clusters predicted either as GM12878 or K562 was shown. In all these analyses, SCATE consistently performed the best.

Second, we conducted peak calling and evaluated each method's ability to recover true peaks in each cell type. Here, the truth was defined as peaks called from the gold standard bulk data, and the evaluation was conducted similar to Fig. 7. Figure 9g and Additional file 2: Figure S8D show the median AUC of all cell clusters predicted either as GM12878 or K562 as a function of cell mixing proportion. SCATE robustly outperformed the other methods.

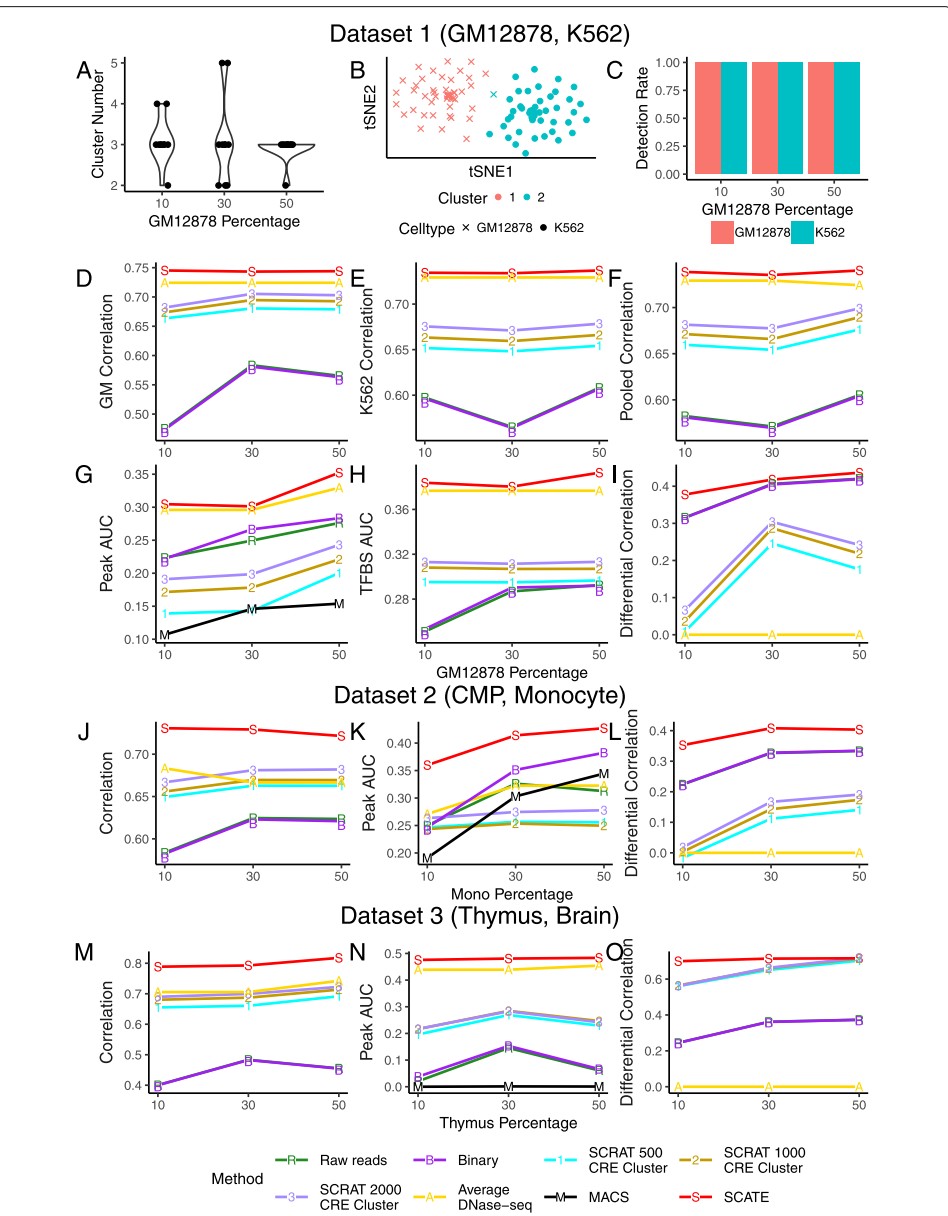

**Fig. 9** Analyses of a heterogeneous cell population. **a** Distribution of cell cluster numbers obtained by SCATE for synthetic samples with different cell mixing proportions. GM12878 and K562 cells are mixed at different proportions. For each mixing proportion, 10 synthetic samples are created and analyzed. **b** An example tSNE plot showing clustering of cells in a synthetic sample. **c** At each cell mixing proportion, the frequency that each cell type is detected in the 10 synthetic samples is shown. **d**–**f** The correlation between the scATAC-seq reconstructed and true bulk regulome in **d** GM12878, **e** K562, and **f** GM12878 and K562 combined for different methods is shown as a function of cell mixing proportion (GM12878 cell percentage). **g** The peak calling AUC (GM12878 and K562 combined) vs. cell mixing proportion. **h** The TFBS prediction AUC (GM12878 and K562 combined) vs. cell mixing proportion. **i** The correlation between the scATAC-seq reconstructed and true bulk differential log-CRE activities is shown as a function of cell mixing proportion. **j**–**l** Similar analyses in samples consisting of human CMP and monocyte cells, including **j** correlation between reconstructed and true bulk log-CRE activities, **k** peak calling AUC, and **l** correlation between predicted and true differential log-CRE activities. **m**–**o** Similar analyses in samples consisting of mouse thymus and brain cells, including **m** correlation between reconstructed and true bulk log-CRE activities, **k** peak calling AUC, and **l** correlation between predicted and true differential log-CRE activities

Third, we compared different methods in terms of their ability to predict TFBSs. TFBS prediction and evaluation were performed similar to Fig. 8. The results are shown in Fig. 9h and Additional file 2: Figure S8E, in which the median AUC for each method is plotted as a function of cell mixing proportion. SCATE produced the best prediction accuracy.

Last but not least, we applied different methods to predict differential CRE activities between different cell types, which is crucial for characterizing the regulatory landscape of a heterogeneous sample. Note that if one views scATAC-seq as a tool for studying cell heterogeneity, then a good analysis method should have the ability to accurately capture differences among cells. Importantly, since differences between cell types are a special case of cell heterogeneity, a good method should be able to keep cell type differences when comparing two cells or two pseudobulk samples from two different cell types. Here, we collected all pairs of cell clusters that were predicted as two different cell types (i.e., one cluster was "predicted GM12878" and the other cluster was "predicted K562"; ambiguous cell clusters were excluded). For each such pair, we computed the difference of reconstructed CRE activities between the two cell clusters. We then compared this predicted difference with the true differential CRE activities derived from the gold standard bulk DNase-seq data for GM12878 and K562. The Pearson correlation between the predicted and true differential signals was calculated. As one example, Fig. 10 and Additional file 2: Figure S9 show the results for a cell cluster pair in a synthetic sample in which 30% of cells was GM12878. SCATE best recovered the differential CRE activities (correlation = 0.43). Figure 9i and Additional file 2: Figure S8F show the median correlation across ten independent synthetic samples at each cell mixing proportion. Once again, SCATE performed the best.

In the above analyses, the Average DNase-seq method completely failed for predicting differential signals between two cell types (correlation = 0) (Figs. 9i, and 10), even though it performed relatively well for estimating CRE activities within one cell type, and peak calling and TFBS prediction in one cell type (Figs. 6, 7, 8, 9f–h). Similarly, each of the

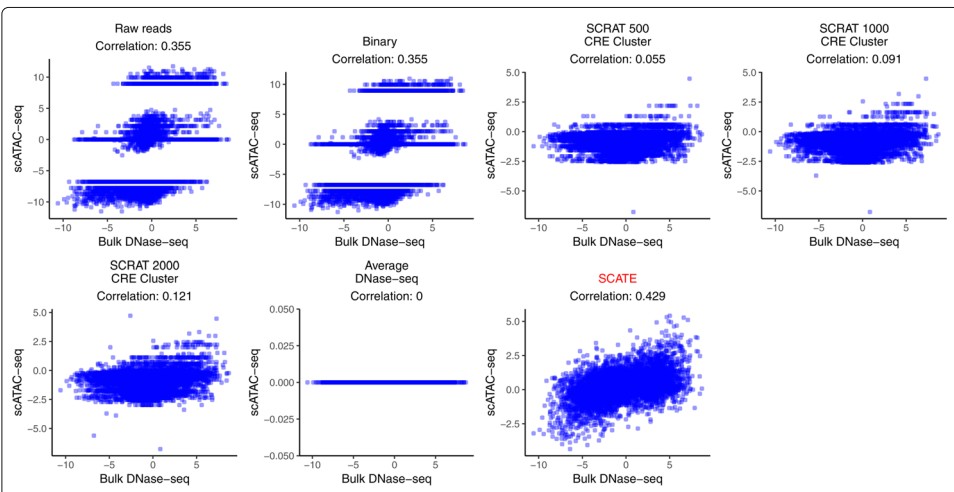

**Fig. 10** An example of predicting differential CRE activities. Scatterplots showing true bulk differential log-CRE activities vs. differential log-CRE activities estimated by different methods in an analysis of a synthetic sample consisting of 30 GM12878 and 70 K562 cells

other methods may perform well in some datasets or analyses but not in others. SCATE is the only method that robustly performed the best in all our analyses.

Similar to GM12878 and K562 (Dataset 1), we also constructed heterogeneous cell populations using the other two datasets (Datasets 2 and 3) and used them to evaluate different methods. The results are shown in Figure 9j–o and Additional file 2: Figures S8G-L and S10. For these two datasets, we did not perform TFBS prediction due to lack of gold standard ChIP-seq data. For estimating CRE activities (Fig. 9j, m, Additional file 2: Fig. S8G,J), peak calling (Fig. 9k, n, Additional file 2: Fig. S8H,K), and predicting differential CRE activities (Fig. 9l, o, Additional file 2: Fig. S8I,L), SCATE again outperformed all the other methods. In many cases, the improvement was substantial (e.g., Fig. 9k, l, n, o, Additional file 2: Fig. S8).

### Example 1: Analysis of scATAC-seq data from human hematopoietic differentiation

To further demonstrate and evaluate SCATE in a more realistic setting, we analyzed a scATAC-seq dataset generated by [27] which consists of 1920 cells from 8 human hematopoietic cell types for which corresponding bulk ATAC-seq data are available. These cell types include hematopoietic stem cell (HSC), multipotent progenitor (MPP), lymphoid-primed multipotent progenitor (LMPP), common myeloid progenitor (CMP), common lymphoid progenitor (CLP), granulocyte-macrophage progenitor (GMP), megakaryocyte-erythrocyte progenitor (MEP), and monocyte (Mono). In this dataset, the true cell type label of each cell was known since cells were obtained by cell sorting. However, they were not used in our SCATE analyses so that our results reflect how data would be analyzed in reality. The true cell type labels were only used after the analysis to evaluate methods. Figure 11a shows the tSNE [32] plot of all cells color-coded by their true cell types. In the plot, different cell types were distributed along three major differentiation lineages (myeloid: HSC→ *MPP* →(CMP or LMPP)→ *GMP* →Mono; erythroid: HSC→ *MPP* → *CMP* →MEP; lymphoid: HSC→ *MPP* → *LMPP* →CLP), which are consistent with known biology. For method evaluation, we analyzed all cells together as a heterogeneous cell population and pretended that the cell type labels were unknown. We also downloaded and processed bulk ATAC-seq data for these 8 cell types from [28] and used them as the gold standard to assess regulome reconstruction accuracy.

Using its default cell clustering method, SCATE identified 14 cell clusters. To evaluate the performance of this unsupervised analysis for recovering true biology, we first assigned a cell type label for each cluster. A cluster was annotated as "predicted cell type X" if the cluster contained at least two cells and the true cell type label of $\geq 70\%$ cells from the cluster was cell type X. Clusters that cannot be annotated using this criterion were labeled as ambiguous. In this way, we were able to unambiguously annotate 9 clusters. Since multiple clusters may be annotated with the same cell type, these 9 annotated clusters corresponded to a total of 6 cell types (Fig. 11b). For these 9 clusters, one can evaluate signal reconstruction accuracy because the bulk ATAC-seq data for the annotated cell type was available. Each cluster was treated as a homogeneous cell population by SCATE and other methods in our analysis (as one would do in real applications), even though the cluster actually may not be pure and may contain cells from more than one cell types. Figure 11d and Additional file 2: Figure S11A compare the Pearson correlation between the gold standard bulk signal and the CRE activities reconstructed from scATAC-seq by different methods. Each boxplot contains 9 data points corresponding to the 9 cell

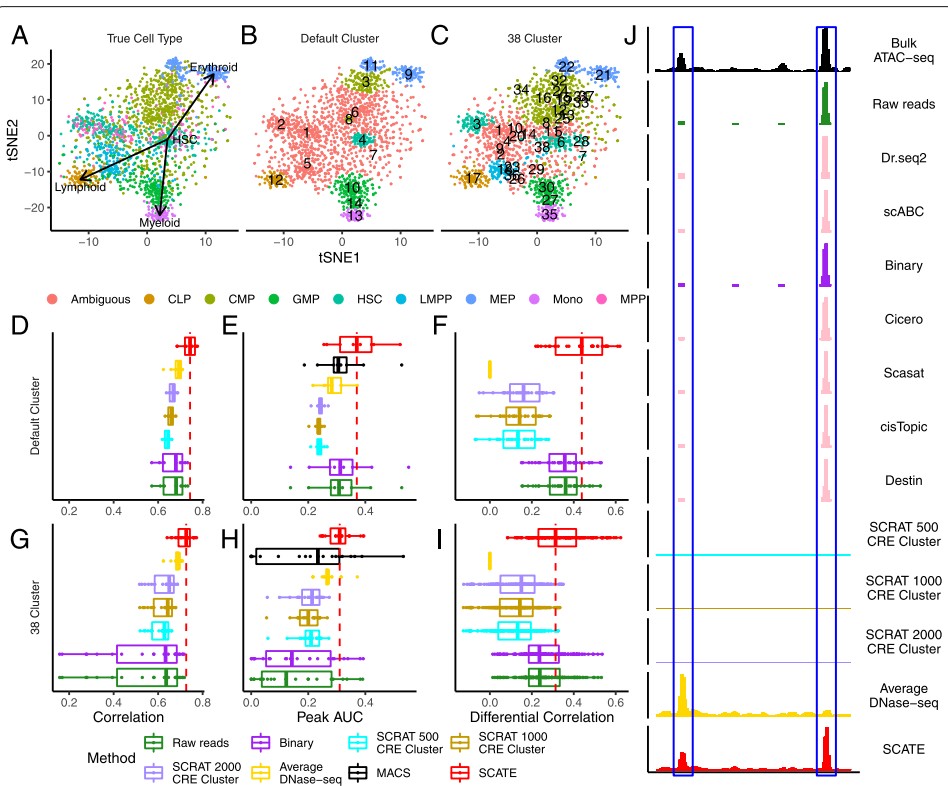

**Fig. 11** Analysis of human hematopoietic differentiation cell types. **a** tSNE plot showing cells color-coded by their true cell types. **b** tSNE plot showing cells color-coded by their predicted cell types. Using the default setting, SCATE grouped cells into 14 clusters (numbers in the plot indicate cluster centers). The clusters that can be unambiguously linked to a cell type are color-coded by cell type. **c** Similar to **b**, but cells are clustered using user-specified cluster number (38 clusters). **d**–**f** Regulome reconstruction performance of different methods in the default analysis, including **d** correlation between reconstructed and true bulk log-CRE activities, **e** peak calling AUC, and **f** correlation between predicted and true differential log-CRE activities. **g**–**i** Regulome reconstruction performance using user-specified cluster number (38 clusters), including **g** correlation between reconstructed and true bulk log-CRE activities, **h** peak calling AUC, and **i** correlation between predicted and true differential log-CRE activities. **j** Comparison of different methods in an example genomic region in HSC cell cluster in the default analysis

clusters. Figure 11e and Additional file 2: Figure S11B compare the peak calling performance (AUC under the sensitivity-FDR curve). Figure 11f and Additional file 2: Figure S11C compare the accuracy for predicting differential CRE activities between different cell types. Here, each data point in the boxplot is a pair of cell clusters annotated with two different cell types. The Pearson correlation between the gold standard bulk differential signal and differential signal reconstructed from scATAC-seq was computed and compared. In all these analyses, SCATE outperformed the other methods. Figure 11j shows an example genomic region in a HSC cell cluster. SCATE most accurately reconstructed the bulk ATAC-seq signal in HSC.

SCATE provides users with the flexibility to specify their own cell cluster number or use their own cell clustering results. The software can reconstruct signals based on user-provided cell cluster number or clustering structure. For instance, suppose one is not satisfied with the default cell clustering and wants to increase the granularity of clustering to make each cluster smaller and more homogeneous, one can manually adjust the cluster number. To demonstrate, we increased the cell cluster number to

38. After increasing the cell cluster number, each cell cluster contained approximately 50 cells on average. After rerunning SCATE, 24 of the 38 cell clusters can be unambiguously annotated, identifying a total of 7 cell types (Fig. 11c). As a comparison, the default analysis only unambiguously identified 6 cell types. For the unambiguously annotated cell clusters, Fig. 11g–i and Additional file 2: Figure S11D-F compare the performance of different methods for reconstructing CRE activities, peak calling, and estimating differential CRE activities between different cell types. SCATE still delivered the best performance. Since the average cell cluster size became smaller, the performance of some methods decreased substantially in some analyses (e.g., the CRE reconstruction and peak calling accuracy for Raw reads and Binary in Fig. 11g, h and Additional file 2: Figure S11D, E). In these cases, the benefit from SCATE was even more obvious.

### Example 2: Analysis of 10x Genomics scATAC-seq data from human peripheral blood mononuclear cells (PBMC)

We also analyzed a scATAC-seq dataset generated from 10x Genomics platform by [33] which consists of 10,027 human peripheral blood mononuclear cells. These cells were sorted into 5 cell types using magnetic-activated cell sorting: B cells, CD4+ T cells, CD8+ T cells, monocytes, and natural killer (NK) cells. Figure 12a visualizes the data with true cell type labels color-coded. Consistent with known biology, CD4+ T cells, CD8+ T cells, and NK cells are closer to each other, whereas B cells and monocytes form more distinct clusters. Again, in order to illustrate how scATAC-seq data would be analyzed in reality, we pretended that cells' true cell type labels are unknown when running SCATE and other methods. We downloaded and processed bulk ATAC-seq data for these cell types from [28] and used them as the gold standard.

Using its default cell clustering method, SCATE identified 15 cell clusters. To evaluate how this unsupervised analysis recovered true biology, we first computationally assigned a cell type label to each cell cluster using the same protocol as in Example 1's hematopoietic analysis. In this way, we were able to unambiguously annotate 11 clusters representing all 5 cell types (Fig. 12b). For these 11 clusters, we evaluated signal reconstruction accuracy using the bulk ATAC-seq data of the annotated cell type. SCATE again outperformed the other methods in terms of Pearson correlation between the gold standard bulk signal and the CRE activities reconstructed from scATAC-seq (Fig. 12d, Additional file 2: Fig.S12A), peak calling performance (Fig. 12e, Additional file 2: Fig. S12B), and accuracy for predicting differential CRE activities between different cell types (Fig. 12f, Additional file 2: Fig. S12C). Figure 12j shows an example genomic region in a B cell cluster. SCATE most accurately reconstructed the bulk ATAC-seq signal in B cells.

In the default analysis, the average number of cells in a cluster for the 11 annotated clusters was 852. Similar to Example 1, we also rerun the analysis by manually setting the cell cluster number to 100 to increase the granularity of clustering. This reduced the average number of cells in a cluster to 100. After running SCATE, we were able to unambiguously annotate 86 clusters corresponding to the 5 cell types (Fig. 12c). Figure 12g–i and Additional file 2: Fig. S12D-F show that SCATE still delivered the best overall performance.

## Discussions

In summary, SCATE provides a new tool for analyzing scATAC-seq data. Our analyses show that it robustly outperforms the existing methods for reconstructing activities of each individual CRE. In many cases, the gain can be substantial.

The main novelty of SCATE is its unique strategy to reconstruct CRE activities from sparse data by (1) integrating data from both similar CREs and cells, (2) leveraging the rich information provided by publicly available regulome data, and (3) adaptively optimizing the analysis resolution based on available data. Coupled with appropriate cell clustering, SCATE allows one to systematically characterize the regulatory landscape of a heterogeneous sample via unsupervised identification of cell subpopulations and reconstruction of their chromatin accessibility profile at the single CRE resolution.

Since many methods for clustering cells using scATAC-seq data have been developed (Additional file 1: Table S1), cell clustering per se is not the focus of this article. In principle, the SCATE model may be coupled with any cell clustering method. While our implementation uses model-based clustering as the default, users are provided with the

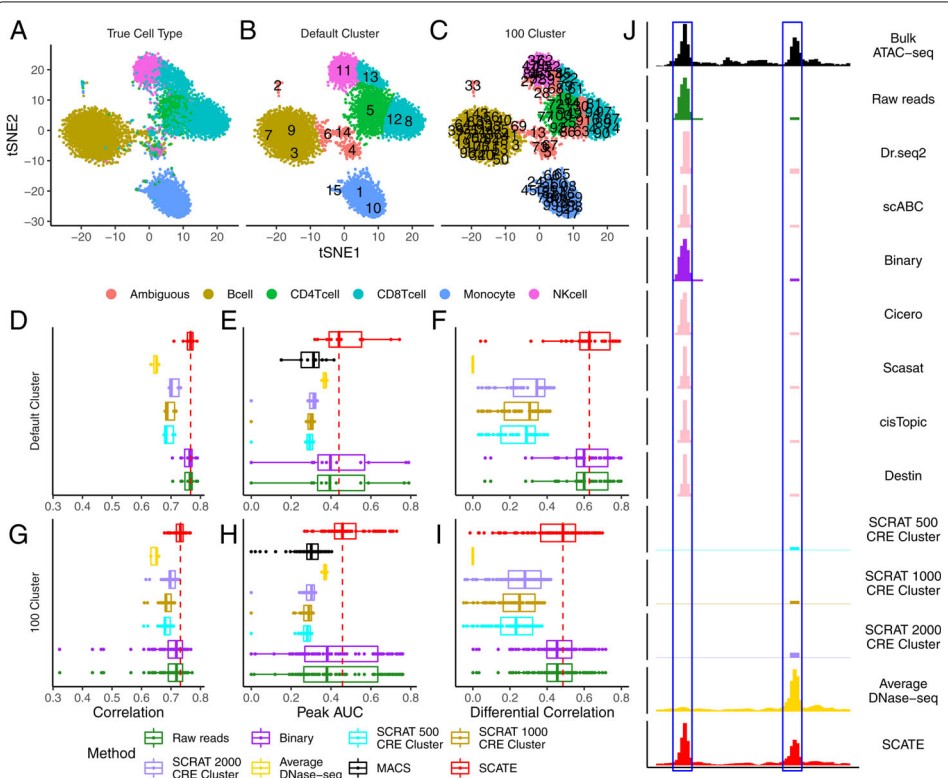

**Fig. 12** Analysis of human PBMCs from the 10x Genomics platform. **a** tSNE plot showing cells color-coded by their true cell types. **b** tSNE plot showing cells color-coded by their predicted cell types. Using the default setting, SCATE grouped cells into 15 clusters (numbers in the plot indicate cluster centers). The clusters that can be unambiguously linked to a cell type are color-coded by cell type. **c** Similar to **b**, but cells are clustered using user-specified cluster number (100 clusters). **d**–**f** Regulome reconstruction performance of different methods in the default analysis, including **d** correlation between reconstructed and true bulk log-CRE activities, **e** peak calling AUC, and **f** correlation between predicted and true differential log-CRE activities. **g**–**i** Regulome reconstruction performance using user-specified cluster number (100 clusters), including **g** correlation between reconstructed and true bulk log-CRE activities, **h** peak calling AUC, and **i** correlation between predicted and true differential log-CRE activities. **j** Comparison of different methods in an example genomic region in B cell cluster in the default analysis

option to use their own cell clustering results as the input for SCATE. For example, cell clustering may be influenced by cell cycle which is not adjusted for in the default clustering method in SCATE. However, if users want to adjust for cell cycle and have performed their own cell clustering to do so, they could replace the default SCATE clustering with their own cell clustering. As another example, some recent studies suggest that distal regulatory elements such as enhancers may be more informative for clustering cells compared to proximal elements such as promoters [21]. Although this information is not currently considered in our default cell clustering algorithm, users have the flexibility to replace the default SCATE cell clustering by cell clustering obtained from other tools (e.g., [21]) that treat distal and proximal regulatory elements differently. Once cell clustering is given, SCATE will apply the same algorithm to estimate activities of all CREs regardless of whether they are proximal or distal. We note that the input CREs for SCATE are compiled from DNase-seq or ATAC-seq data which cover both proximal and distal elements. When we use these data to detect CREs, proximal and distal elements were not treated differently.

For estimating CRE activities, the default setting of SCATE takes a precompiled list of CREs and their clustering structure as input. These precompiled CREs and clusters are learned from a large number of DNase-seq samples representing diverse cell types in BDDB. As Additional file 2: Figure S1 shows, the precompiled CREs in BDDB typically cover most of the CREs one would detect in a new dataset. The precompiled CRE clusters contain information about which CREs are correlated. The correlation itself does not tell one the actual activity of each CRE in a new scATAC-seq dataset. For example, knowing that CRE X, CRE Y, and CRE Z are correlated does not tell one whether they will have high activity or low activity in a new dataset. To infer CREs' actual activity in scATAC-seq, one also needs to use the read count information from the scATAC-seq data. In SCATE, the prior information learned from BDDB about CRE correlations is combined with the observed read counts in scATAC-seq data to infer CRE activities. In this way, information about how CREs are correlated (but not about the actual activities of CREs) are transferred from the existing BDDB data to the new scATAC-seq data. The transferred correlation information is helpful for improving the estimation of CRE activities. For example, consider two scenarios: (1) CRE Z has 0 read, but all other CREs in the same cluster have non-zero read counts; (2) CRE Z has 0 read, and all other CREs in the same cluster have 0 read. Based on the knowledge that CREs in the same cluster tend to be co-activated, one can infer that CRE Z is more likely to be active in scenario (1) than in scenario (2). In other words, based on the read counts observed at the correlated CREs, the zero read count for CRE Z in scenario (1) is more likely to represent an inaccurate measurement, whereas the zero read count for CRE Z in scenario (2) more likely reflects its real low activity level.

A potential limitation of using our precompiled CRE list and clusters is that for a given version of BDDB, these lists will be fixed and remain the same for analyzing all new scATAC-seq datasets. A new scATAC-seq dataset may contain new CREs and new CRE correlation structures that may not be fully captured by our precompiled CRE list and clusters. For this reason, SCATE also provides functions to support users to compile their own CRE list and CRE clusters. Users can use these functions in two ways. In one way denoted as "SCATE(User Data)," one can compile CREs from their own scATAC-seq data (by clustering cells and detecting CREs in each cell cluster) and cluster CREs based on

their own scATAC-seq data (using normalized CRE-read count matrix, where each row is a CRE and each column is a pseudobulk sample obtained by pooling cells in a cluster). In another way denoted as "SCATE(BDDB+User Data)," users can cluster cells into pseudobulk samples and add the pseudobulk samples obtained from their scATAC-seq data to BDDB to expand the database. One can then compile CREs and their clustering using the expanded BDDB. The difference among the default SCATE, SCATE(User Data) and SCATE(BDDB+User Data) is that (1) the default mode only uses existing data in BDDB to compile input CREs (thus it is also denoted as SCATE(BDDB)), (2) SCATE(User Data) does not use any information from BDDB, and (3) SCATE(BDDB+User Data) combines BDDB with users' own data to compile CREs and hence uses both sources of information.

While SCATE(User Data) and SCATE(BDDB+User Data) provide users with the flexibility to compile dataset-specific CREs, we choose SCATE(BDDB) as the default mode of SCATE for two reasons. First, based on our own experience with real data, SCATE(BDDB) usually performs better than SCATE(User Data) (Additional file 2: Fig. S13). This is likely because CREs and their clustering patterns compiled from diverse cell types in BDDB are more informative than those compiled from a limited number of cell types in a new scATAC-seq data. Second, SCATE(BDDB) and SCATE(BDDB+User Data) usually show similar performance, with SCATE(BDDB+User Data) being slightly better. Despite the slight loss of accuracy, SCATE(BDDB) is substantially easier to use. In order to use SCATE(BDDB+User Data), users have to download the DNase-seq data in BDDB and run CRE detection and clustering themselves which require extensive computation (it typically takes 1–2 days in a computer with 20 cores (2.5 GHz CPU/core)). By contrast, in order to use SCATE(BDDB), one can skip these tedious and computation heavy steps and only download the precompiled CRE list and clustering. With these precompiled CREs and clusters, running SCATE only takes a few minutes per cell cluster.

In the future, the SCATE framework may be extended in multiple directions. For example, how should one account for the effects of cell cycles in scATAC-seq analysis remains an open problem. Addressing this problem requires robust methods to accurately infer cells' phase in cell cycle using scATAC-seq data and systematic benchmark datasets and method evaluation. Both are non-trivial for scATAC-seq and are beyond the scope of this study. However, they are interesting topics for future research. As another example, our current implementation of SCATE is focused on identifying and characterizing cell subpopulations. A future direction is to extend this framework to other types of analyses such as pseudotime analysis [34] to allow the study of CRE activities along continuous pseudotemporal trajectories. In the future, it is also useful to develop new methods that utilize the improved CRE estimation to more accurately reconstruct gene regulatory networks.

The basic framework adopted by SCATE to improve the analysis of sparse data by integrating multiple sources of information is general. In principle, a similar approach may also be used to analyze other types of single-cell epigenomic data such as single-cell DNase-seq or ChIP-seq, and possibly single-cell Hi-C [35].

## Methods
### Single-cell ATAC-seq data preprocessing
Single-cell ATAC-seq data for GM12878 and K562 cells were obtained from GEO (GSE65360) [4]; single-cell ATAC-seq data for human hematopoietic cell types were obtained from GEO (GSE96769) [27]; single-cell ATAC-seq data for mouse brain and

thymus were obtained from GEO (GSE111586) [29]. For each cell, paired-end reads were trimmed using the program provided by [4] to remove adaptor sequences. Reads were then aligned to human (hg19) or mouse (mm10) genome using bowtie2 with parameter -X2000. This parameter retains paired reads with insertion up to 2000 base pairs (bps). PCR duplicates were removed using Picard (http://broadinstitute.github.io/picard/).

The 10x Genomics single-cell ATAC-seq data for human PBMC were downloaded from GEO (GSE129785) [33]. The 10x Cell Ranger ATAC Software Suite were used to process reads and align them to human hg19 genome. All other analysis procedures were the same as the analysis of human hematopoietic cell types.

### Genome segmentation

Genome is segmented into 200 base pair (bp) nonoverlapping bins. Bins that overlap with ENCODE blacklist regions are excluded from subsequent analyses since their signals tend to be artifacts [36].

### Bulk DNase-seq database (BDDB)

SCATE borrows information from large amounts of publicly available bulk DNase-seq data to improve scATAC-seq analysis. We compiled a database consisting of 404 human and 85 mouse DNase-seq samples obtained from the ENCODE. Take human as an example, we downloaded all ENCODE DNase-seq samples generated by the University of Washington [15] in bam format. Files marked by ENCODE as low quality (marked as "extremely low spot score" or "extremely low read depth" by ENCODE) were filtered out. Technical replicates for each distinct cell type or tissue were merged into one sample. This has resulted in 404 DNase-seq samples representing diverse cell types (Additional file 4: Table S3). Mouse samples were processed similarly (Additional file 5: Table S4).

### Compiling cis-regulatory elements (CREs) using bulk data compendium

Given a species and a compendium of bulk regulome samples (e.g., DNase-seq samples in BDDB), SCATE systematically identifies CREs in the genome as follows. Let $y_{i,j}$ denote the raw read count of bin $i$ in sample $j$. Let $L_j$ be sample $j$'s total read count divided by $10^8$ (i.e., the library size in the unit of hundred million. For example, a sample with 200 million reads has $L_j = 2$). We normalize the raw read counts by library size and log2-transform them after adding a pseudocount 1. This results in normalized data $\tilde{y}_{i,j} = \log_2(y_{i,j}/L_j + 1)$. Bin $i$ is called a "signal bin" in sample $j$ if (1) $y_{i,j} \geq 10$, (2) $\tilde{y}_{i,j} \geq 5$, and (3) $\tilde{y}_{i,j}$ is at least five times (three times for mouse) larger than the background signal defined as the mean of $\tilde{y}_{i,j}$s in the surrounding 100 kb region. The cutoffs for defining signal bins are used to filter out noisy genomic loci since including such loci will increase computational burden. For example, the CRE clustering below failed to run on our computer when we included all genomic bins in the analysis. We explored different choices of cutoffs that were computationally feasible on our computer and found that the cutoffs used above had good empirical performance compared to using looser or more stringent cutoffs (see details in Additional file 2: Fig. S14 and Additional file 6: Supplementary Note).

If a bin is a signal bin in at least one bulk sample, it is labeled as a "known CRE." In this way, all genomic bins are labeled as either "known CREs" or "other bins." 522,173 known

CREs for human and 475,865 known CREs for mouse are identified using our bulk DNase-seq compendium. Locations of these CREs are stored in SCATE and provided as part of the software package. Saturation analysis shows that typically a new bulk sample from a new cell type only contributes a small fraction (0.013% for human and 0.18% for mouse) of new CREs to the known CRE list (Additional file 2: Fig. S1A). In the three benchmark scATAC-seq datasets used in this article, datasets 1, 2, and 3 would only add 0.050%, 0.0013%, and 0.063% new CREs, respectively, to our known CRE list. For the human hematopoietic differentiation and PBMC datasets used in the last two "Results" sections, the scATAC-seq dataset would only add 0.118% and 0.058% of new CREs to the known CRE list, respectively (Additional file 2: Fig. S1B; the calculation was based on detecting CREs in each cell type separately and then adding the union of all CREs from all cell types in the scATAC-seq data to the known CRE list). This suggests that the majority of a new sample's regulome can be studied by analyzing the precompiled known CREs, which can save user's work on compiling and clustering their own CREs. In this article, SCATE is demonstrated using our precompiled known CRE list, as the performance curves and statistics do not change much by adding new CREs from each scATAC-seq dataset to the analysis.

### SCATE model for known CREs in a single cell

Consider scATAC-seq data from one single cell $j$. Given aligned sequence reads, SCATE will estimate activities of known CREs first. Let $y_{i,j}$ denote the observed read count for CRE $i$ ($i = 1, \ldots, I$) in cell $j$, and let $\mu_{i,j}$ denote the unobserved true activity. Our goal is to infer the unobserved $\mu_{i,j}$ from the observed data $y_{i,j}$. We assume the following data generative model:

$$
\begin{aligned}
y_{i,j} &\sim Poisson(L_j \mu_{i,j}^{sc}) \\
\log(\mu_{i,j}^{sc}) &= h_j(\log(\mu_{i,j})) \\
\log(\mu_{i,j}) &= m_i + s_i \delta_{i,j} \\
\boldsymbol{\delta}_j &= \mathbf{X}\boldsymbol{\beta}_j
\end{aligned}
\tag{1}
$$

This model has three main components which will be explained below.

1. *Model for true activity.* The unobserved $\mu_{i,j}$ is modeled as $\log(\mu_{i,j}) = m_i + s_i \delta_{i,j}$. Here $m_i$ and $s_i$ represent CRE $i$'s baseline mean activity and standard deviation (SD). They are used to model the locus-specific but cell-type-independent baseline behavior of each CRE (i.e., the locus effects observed in Fig. 1e). Since these locus-specific effects cannot be reliably learned using sparse data or data from one cell type, we learn them using the bulk data from diverse cell types in our bulk regulome data compendium (see below). Once they are learned, $m_i$ and $s_i$ are treated as known. The unknown $\delta_{i,j}$ describes CRE $i$'s cell-specific activity after removing locus effects (i.e., $\delta_{i,j} = \frac{\log(\mu_{i,j}) - m_i}{s_i}$). Due to data sparsity, accurately estimating $\delta_{i,j}$ using the observed data from only one CRE in one cell is difficult. Thus, we impose additional structure on $\delta_{i,j}$s to allow co-activated CREs to share information to improve the estimation. We group CREs into $K$ clusters based on their co-activation patterns across cell types (see below). We assume that CREs in the same cluster share the same $\delta$. Mathematically, let $\boldsymbol{\delta}_j = (\delta_{1,j}, \ldots, \delta_{I,j})^T$ be a column vector that contains $\delta_{i,j}$s from all CREs in cell $j$. Let $\mathbf{X}$ be a $I \times K$ cluster

membership matrix. Each entry of this matrix $x_{ik}$ is a binary variable: $x_{ik} = 1$ if CRE $i$ belongs to cluster $k$, and $x_{ik} = 0$ otherwise. Let $\beta_{k,j}$ denote the common activity of all CREs in cluster $k$. Arrange $\beta_{k,j}$s into a column vector $\boldsymbol{\beta}_j = (\beta_{1,j}, \ldots, \beta_{K,j})^T$. Our assumption can be represented as $\boldsymbol{\delta}_j = \mathbf{X}\boldsymbol{\beta}_j$. When the cluster number $K$ is smaller than the CRE number $I$, imposing this additional structure on $\delta_{i,j}$ reduces the number of unknown parameters from $I$ to $K$. As a result, it increases the average amount of information available for estimating each parameter. Note that in our model, two CREs with the same $\delta$ can still have different activities (i.e., different $\mu_{i,j}$s) because $\log(\mu_{i,j}) = m_i + s_i\delta_{i,j}$. In other words, SCATE allows co-activated CREs to share information through $\delta$, but at the same time it also allows each CRE to keep its own locus-specific baseline characteristics. This is an important feature missing in other existing methods. Another unique feature of SCATE is that we treat the cluster number $K$ as a tuning parameter and adaptively choose it based on available information to optimize the analysis' spatial resolution. Unlike SCATE, other existing methods aggregate CREs based on known pathways. For them, $K$ is fixed and the analysis' spatial resolution cannot be tuned and optimized.

2. *Model for technical bias.* Since the locus effects $m_i$ and $s_i$ are learned from the bulk data, we view $\mu_{i,j}$ as the activity one would obtain if one could measure a bulk regulome sample (e.g., bulk DNase-seq) consisting of cells identical to cell $j$. In scATAC-seq data, $\mu_{i,j}$ is distorted to become $\mu_{i,j}^{sc}$ due to technical biases in single-cell experiments (e.g., DNA amplification bias). We model these unknown technical biases using a cell-specific monotone function $h_j(.)$. In other words, we assume $\log(\mu_{i,j}^{sc}) = h_j(\log(\mu_{i,j}))$. We estimate the unknown function $h_j(.)$ by comparing scATAC-seq data with the bulk regulome data at CREs that show constant activity across different cell types (see below). Once $h_j(.)$ is estimated, it is assumed to be known.

3. *Model for observed read counts.* We assume that the observed read count $y_{i,j}$ is generated from a Poisson distribution with mean $L_j\mu_{i,j}^{sc}$. Here $L_j$ is the total number of reads in cell $j$ divided by $10^8$. It is a cell-specific normalizing factor to adjust for library size.

For a fixed cluster number $K$, we fit the model as follows: (1) use the bulk regulome data compendium to learn locus effects $m_i$ and $s_i$; (2) use scATAC-seq data and the bulk regulome data compendium to learn technical bias function $h_j(.)$ which normalizes scATAC-seq data with the bulk regulome compendium used to learn locus effects; (3) given $m_i$, $s_i$ and $h_j(.)$, use the observed data $\mathbf{y}$ to estimate $\boldsymbol{\beta}$ which will determine $\boldsymbol{\delta}$ and $\boldsymbol{\mu}$. The estimated $\boldsymbol{\mu}$ provides the final estimates for CRE activities.

In order to optimize the analysis' spatial resolution, SCATE treats the cluster number $K$ as a tuning parameter. CREs are clustered at multiple granularity levels corresponding to different $K$s. As $K$ increases, the average number of CREs per cluster decreases. This increases spatial resolution because the cluster activity more resembles the activity of individual CREs. However, increasing $K$ also decreases the amount of information for estimating the activity of each cluster, and thus the estimates become noisier. We use a cross-validation approach to choose the optimal $K$ that balances spatial resolution and estimation uncertainty (see below).

**Estimate locus effects $m_i$ and $s_i$**

We estimate locus effects using the rich bulk data from diverse cell types in the bulk regulome compendium. Let $y_{i,j}$ be the observed read count for genomic bin $i$ and bulk sample $j$ $(j = 1, \ldots, J)$. $L_j$ represents sample $j$'s library size in the unit of hundred million. For each genomic bin $i$, locus effects are estimated using the observed counts $\{y_{i,j} : j = 1, \ldots, J\}$. We model $y_{i,j}$ in bulk data as:

$$y_{i,j} \sim Poisson(L_j \mu_{i,j})$$
$$\log(\mu_{i,j}) = m_i + s_i \delta_{i,j} \tag{2}$$

This is similar to the single-cell model above but without the technical bias component. Without additional constraints, $m_i$ and $s_i$ are not identifiable since each bin $i$ has only $J$ observed data points but $J+2$ unknown parameters (i.e., $m_i$, $s_i$, and $J$ different $\delta_{i,j}$s). Thus, we further assume $\delta_{i,j} \sim N(0, 1)$. This is equivalent to assuming that $\log(\mu_{i,j})$ for bin $i$ is normally distributed, and $m_i$ and $s_i$ are its mean and SD respectively. This assumption is based on observing that CREs' log-normalized read counts after standardization (i.e. subtract $m_i$ and divide by $s_i$) are approximately normally distributed (Additional file 2: Fig. S15). With this additional constraint, $m_i$ and $s_i$ become identifiable. Since maximum likelihood estimation for all genomic bins in a big genome like human is computationally slow, SCATE employs the method of moments to estimate $m_i$ and $s_i$. Based on the model and theoretical moments of Poisson and Lognormal distributions, the first and second moments of $y_{i,j}/L_j$ are (see Additional file 7: Supplemental Note for derivations):

$$E\left(\frac{y_{i,j}}{L_j}\right) = e^{m_i + \frac{1}{2}s_i^2}$$
$$E\left(\frac{y_{i,j}}{L_j}\right)^2 = \frac{1}{L_j} e^{m_i + \frac{1}{2}s_i^2} + \left[e^{m_i + \frac{1}{2}s_i^2}\right]^2 e^{s_i^2} \tag{3}$$

By matching the model-based moments to the empirical first two moments of the observed $y_{i,j}/L_j$s, we obtain the following closed-form estimates for $m_i$ and $s_i$ which can be computed efficiently:

$$\tilde{s}_i = \sqrt{\log\left(\frac{\sum_j (y_{i,j}/L_j)^2/J - \sum_j (y_{i,j}/L_j^2)/J}{(\sum_j (y_{i,j}/L_j)/J)^2}\right)}$$
$$\tilde{m}_i = \log\left(\frac{\sum_j (y_{i,j}/L_j)}{J}\right) - \tilde{s}_i^2/2 \tag{4}$$

In rare cases where $\frac{\sum_j (y_{i,j}/L_j)^2/J - \sum_j (y_{i,j}/L_j^2)/J}{(\sum_j (y_{i,j}/L_j)/J)^2} < 1$, the estimates become:

$$\tilde{s}_i = 0$$
$$\tilde{m}_i = \log\left(\frac{\sum_j (y_{i,j}/L_j)}{J}\right) \tag{5}$$

**Estimate technical bias function $h_j(.)$**

The cell-specific technical bias function $h_j(.)$ is estimated using known CREs whose activities do not change much across cell types. For each CRE, the $\tilde{s}_i$ estimated above reflects its variability across diverse cell types in the bulk regulome data compendium. To select low-variability CREs, we first group all known CREs into 10 strata based on their baseline mean activity values (i.e., $\tilde{m}_i$s). To do so, the $\tilde{m}_i$s from all CREs are collected and their

10%, 20%, ..., 90% quantiles are computed. These quantiles are used to define the 10 strata. Within each stratum, we find 1000 CREs with the smallest $\tilde{s}_i$ values. The union set of these 10000 CREs creates the set $\mathcal{H}$ of "low-variability" CREs. For these low-variability CREs, their activities are almost constant across cell types. Thus, one can assume that their activities in a new cell are known and approximately equal to $\tilde{m}_i$, and the model for their scATAC-seq read counts in a new cell $j$ can be simplified to:

$$
\begin{aligned}
y_{i,j} &\sim Poisson(L_j \mu_{i,j}^{sc}) \\
\log(\mu_{i,j}^{sc}) &= h_j(\log(\mu_{i,j})) \approx h_j(\tilde{m}_i)
\end{aligned}
\tag{6}
$$

We estimate $h_j(.)$ using $y_{i,j}$s from these low-variability CREs. The function $h_j(.)$ is monotonically increasing but has unknown form. We model it using monotone spline [37] (splines2 package in R):

$$
h_j(x) = \alpha_{j,0} + \sum_{t=1}^{T} \alpha_{j,t} I_t(x) \quad s.t. \; \alpha_{j,t} \geq 0 \; (t = 1, ..., T)
$$

Here, $I_t(x)$ are known I-spline basis functions (which are monotone functions [37]) and $\alpha_{j,t}$s are unknown regression coefficients. The constraints $\alpha_{j,t} \geq 0$ make $h_j(.)$ monotone and non-decreasing. The maximum likelihood estimates for coefficients $\boldsymbol{\alpha}_j = \{\alpha_{j,t} : t = 0, \ldots, T\}$ can then be obtained as:

$$
\tilde{\boldsymbol{\alpha}}_j = \underset{\boldsymbol{\alpha}_j}{\arg\max} \sum_{i \in \mathcal{H}} [y_{i,j} * h(\tilde{m}_i) - L_j e^{h(\tilde{m}_i)}] \quad s.t. \; \alpha_{j,t} \geq 0 \; (t = 1, ..., T)
\tag{7}
$$

To select the optimal set of basis functions, we try different settings of knots by changing $T$. We set $T = 1, 2, ..., 6$, respectively, which sets the number of knots from 0 to 5. For each $T$, the $t/T$th quantiles ($t = 1, ..., T-1$) of $\tilde{m}_i$ are chosen as the knots. Given the knots, the spline basis functions are then generated by splines2. The $T$ with the smallest Bayesian information criterion (BIC) is chosen to obtain the optimal set of basis functions.

**Estimate $\beta$, $\delta$, and $\mu$**

Once the locus effects $m_i$ and $s_i$ and technical bias function $h_j(.)$ are estimated, SCATE treats them as known and will then estimate $\boldsymbol{\beta}$. Suppose CREs are grouped into $K$ clusters. The activity for cluster $k$ in cell $j$, $\beta_{k,j}$, can be estimated using the observed read counts in cell $j$ for all CREs in the cluster. When data are sparse (particularly for clusters with small number of CREs), the maximum likelihood estimate can be unreliable due to its high variance. Thus, consistent with our bulk regulome data model, we impose a prior distribution on $\beta_{k,j}$ to help regularize its estimation: $\beta_{k,j} \sim N(0, 1)$. We then estimate $\beta_{k,j}$ using its posterior mode:

$$
\tilde{\beta}_{k,j} = \underset{\beta}{\arg\max} \sum_{i \in C(k)} \left[ y_{i,j} h_j(m_i + s_i \beta) - L_j e^{h_j(m_i + s_i \beta)} \right] - \beta^2/2
$$

Here, $C(k)$ represents the set of CREs in cluster $k$. The above optimization involves only one variable $\beta$, and thus the computation is not expensive. Estimation of different $\beta_{k,j}$s are handled separately.

Given $\tilde{\beta}_{k,j}$, $\delta_{i,j}$ and $\mu_{i,j}$ can be derived using model (1).

**Analysis at multiple spatial resolution levels (i.e., multiple $K$s)**

SCATE analyzes data at multiple spatial resolution levels by setting the cluster number $K$ to different values. To do so, known CREs are clustered based on their co-activation

patterns across all samples in the bulk regulome data compendium. Before clustering, CREs' normalized data $\tilde{y}_{i,j}$ are organized as a matrix. Rows of the matrix correspond to CREs and columns correspond to samples. Each row is standardized to have zero mean and unit SD. Then, CREs (i.e., rows) are clustered hierarchically at multiple granularity levels. A naive hierarchical clustering of 522,173 CREs (475,865 CREs for mouse) is difficult because it requires computing a distance matrix on the order of $500,000 \times 500,000$. To make the computation tractable, SCATE employs a three-stage clustering approach.

- Stage 1: K-means clustering (Euclidean distance) is used to group all CREs into 5000 clusters. Each cluster consists of a group of CREs with similar cross-sample activity patterns. The mean number of CREs contained in each cluster is approximately 100 (for human 522,173 CREs/5000 clusters = 104 CREs/cluster; for mouse 475,865 CREs/5000 clusters = 95 CREs/cluster). The end product of this stage is 5000 CRE clusters. For each cluster, the mean activity of all CREs in each sample is computed. It is then standardized to have zero mean and unit SD across samples.
- Stage 2: To obtain coarser clusters, the 5000 clusters from stage 1 are grouped hierarchically using hierarchical clustering (Euclidean distance, complete agglomeration) based on their mean activity profile. In this way, CREs are hierarchically grouped into 5000, 2500, 1250, 625, 312, and 156 clusters.
- Stage 3: To obtain fine-grained clusters, for each cluster obtained in Stage 1, hierarchical clustering is applied to split CREs in that cluster into smaller clusters. In this way, each cluster from Stage 1 can be divided into 2, 4, 8, ... subclusters until each subcluster contains only one CRE. For different Stage 1 clusters, their CRE numbers are different and therefore the exact number of their subclusters obtained in Stage 3 may vary.

After all three stages, we obtain clusters of CREs at multiple granularity levels. In other words, CREs are grouped into $K$ clusters for different $K$ values. For human, $K = 156, 312, 625, 1250, 2500, 5000, 9856, 19008, 35361, 64398, 117596, 213432, 521820$. For mouse, $K = 156, 312, 625, 1250, 2500, 5000, 9996, 19953, 39732, 78868, 154813, 283422, 465055$. CREs' clustering structure for human and mouse obtained using our BDDB DNase-seq compendium is stored and provided as part of the SCATE package. Users can use it directly without recomputing them.

### Optimizing spatial resolution (*K*) by cross-validation

SCATE optimizes the spatial resolution of the analysis by choosing the optimal $K$ via cross-validation. For a given $K$, after clustering CREs, CREs are randomly partitioned into a training set (90% CREs) and a testing set (10% CREs). Next, for each cluster $k$, CREs in the training set are used to estimate $\beta_{k,j}$ which is the common activity of all CREs in that cluster. Using the estimated $\tilde{\beta}_{k,j}$, the log-likelihood of the test CREs in cluster $k$ can be computed according to model (1) because they share the same $\beta_{k,j}$ with training CREs in the same cluster. We perform the same calculations for all clusters and obtain the median log-likelihood of all testing CREs.

The above procedure is run for different values of $K$. The cluster number $K$ with the largest median log-likelihood in test data is selected as the optimal $K$.

### Postprocessing – SCATE for other genomic bins in a single cell

After estimating activities of known CREs, SCATE will analyze all other bins in the genome. These bins fall into two classes. First, some bins have zero scATAC-seq read count across all cells. For these bins, $\mu_{i,j}$ is estimated to be zero. Second, the remaining bins have at least one read in the scATAC-seq data. For these bins, we estimate $\mu_{i,j}$ using a predictive machine learning approach xgboost (eXtreme Gradient Boosting [38]) where the response variable is the SCATE signal $\tilde{\mu}_{i,j}$ and the predictors are normalized read count $y_{i,j}/L_j$, $m_i$ and $s_i$. The model is trained using known CREs. The trained model is then applied to bins not included in the known CRE list to make predictions. This will transform the read counts at these bins to a scale consistent with the reconstructed activities for known CREs.

### SCATE for multiple cells

When a scATAC-seq dataset contains multiple cells, we first cluster cells using a method similar to our previously published method SCRAT [13]. Before clustering cells, CREs are grouped into 5000 clusters using BDDB as before. For each cell, the average activity of all CREs in each CRE cluster is calculated as in SCRAT. This transforms the scATAC-seq data in each cell into a feature vector consisting of 5000 CRE cluster activities. After quantile normalizing features across cells, features with low-variability across cells are filtered out. To identify low-variability features, for each feature, we calculate the mean and SD of its activity across cells. Using the means and SDs of all features, we fit a polynomial regression with degree=3 to describe the relationship between the SD (response) and mean (independent variable). Features for which the observed SD is smaller than the expected SD (from the fitted model) given the mean activity are filtered out. Among the remaining high-variability features, we retain those that have non-zero read count in at least 10% of cells. PCA is then performed on the retained features. The top 50 principal components are then used to perform tSNE. The model-based clustering (mclust in R) [26] is used to perform clustering on tSNE space with default settings. The cluster number is chosen based on the Bayesian Information Criterion in mclust. If users do not want to use the default cluster number or clustering method, SCATE also provides an option to allow them to specify the cluster number by their own or use their own clustering results from other algorithms.

After cell clustering, each cluster consists of a set of similar cells and represents a relatively homogeneous cell subpopulation. SCATE will estimate the regulome profile of each cluster. For each cell cluster, reads from all cells are pooled together to create a pseudo-cell. The SCATE model for a single cell described above is then applied to the pseudo-cell to estimate CRE activities. For instance, the bias normalizing function $h_j(.)$ is estimated by treating the pseudo-cell obtained from cluster $j$ (after pooling cells) as a single cell. The estimated regulome profile of the pooled sample typically will achieve higher spatial resolution than a single cell since (1) the pseudo-cell contains data from more than one cell and (2) SCATE automatically tunes the spatial resolution based on available information. The output of SCATE is the estimated regulome profile for each cell subpopulation.

### Peak calling and evaluation

A moving average approach is used to call peaks from the reconstructed regulome profile. Given a moving window size $2W + 1$, the moving average signal for each 200 bp bin is

calculated as the average signal of the bin and its $2W$ neighboring bins ($W$ bins on the left and $W$ bins on the right). By default, $W = 1$ which amounts to averaging signals from 3 bins spanning 600 bp in total. In parallel, we also calculate the average signal of $2W + 1$ randomly selected bins (not necessarily neighboring bins) for 100,000 times to construct a background distribution for the moving average signal. For a genomic bin with moving average signal $s$, the false discovery rate (FDR) is estimated as the proportion of background distribution larger than $s$ divided by the observed proportion of genomic bins with signals larger than $s$. Genomic bins with FDR smaller than 0.05 are identified and consecutive bins are merged into peaks. Peaks are ranked by FDR. For peaks tied with the same FDR, they are ranked further by the moving average signals.

For evaluation, peaks called using signals constructed by different methods are compared with peaks called using bulk regulome data. In the evaluation, we also assessed MACS peak calling on pooled cells. MACS is run with settings –nomodel –extsize 147.

### TFBS prediction

TF motifs are downloaded from JASPAR [39] (Additional file 3: Table S2). These motifs were mapped to the genome using CisGenome with likelihood ratio cutoff = 100. Narrow peak files of the corresponding ChIP-seq data in GM12878 and K562 are downloaded from ENCODE. For each TF and cell type, genomic bins with motif were ranked based on reconstructed scATAC-seq signals to predict TFBSs. Genomic bins with motif that overlap with ChIP-seq peaks are used as gold standard.

### Processing of benchmark bulk DNase-seq and ATAC-seq data

The benchmark bulk DNase-seq data for GM12878 and K562 (Dataset 1) are obtained from ENCODE. Bulk ATAC-seq data for human CMP and monocytes (Dataset 2), human hematopoietic cell types, and human PBMC in the last two examples are obtained from GEO under accession GSE74912. Bulk DNase-seq data for mouse brain and thymus (Dataset 3) are obtained from ENCODE.

Bulk DNase-seq samples are processed using the same protocol as DNase-seq data processing in BDDB. For ATAC-seq sample, reads are aligned to human genome hg19 using bowtie with parameters (-X 2000 -m 1). PCR duplicates are removed by Picard (http://broadinstitute.github.io/picard/). The aligned reads are used to obtain bin read counts.

### BDDB data variance explained by mean DNase-seq profile

Denote $\tilde{y}_{i,j}$ as the log-normalized read count for CRE $i$ ($i = 1, 2, ..., I$) and sample $j$ ($j = 1, 2, ..., J$) in BDDB. Denote $a_i$ as the mean DNase-seq activity (i.e., mean of $\tilde{y}_{i,j}$s) for CRE $i$ computed using BDDB samples. Denote $\bar{\bar{y}} = \sum_{ij} \tilde{y}_{i,j}/(IJ)$, and $\bar{a} = \sum_i a_i/I$. The proportion of variance explained is calculated as $J \sum_i (a_i - \bar{a})^2 / \sum_{ij} (\tilde{y}_{i,j} - \bar{\bar{y}})^2$.

### Analyses using existing methods

Existing methods were run using their default parameter settings implemented in their software or reported in their original publications. All these methods used MACS for peak calling. The parameters for running MACS as reported in their original papers are:

- Cicero: MACS2 with parameters –nomodel –extsize 200 –shift -100 –keep-dup all.
- Dr.seq2: MACS1.4 with parameters –keep-dup 1 –nomodel –shiftsize 73.

- Scasat: MACS2 with parameters –nomodel, –nolambda, –keep-dup all –call-summits -p 0.0001.
- scABC: MACS2 with parameters -p 0.1.
- cisTopic: MACS2 with parameters –nomodel -q 0.001.
- Destin: MACS2 with parameters –nomodel -p 0.01.

## Software

SCATE is implemented as an R package. It can be installed from GitHub. In terms of computational time, compiling CREs and clustering CREs typically take 1–2 days. Given the CRE list and CREs' clustering structure, running SCATE to reconstruct regulome approximately takes 5 minutes per cell cluster on a computer with 10 computing cores (2.5 GHz CPU/core) and a total of 20GB RAM.

## Supplementary information

---

**Additional file 1:** Table S1. A comparison between SCATE and other existing methods.

**Additional file 2:** Figures S1-S15.

**Additional file 3:** Table S2. List of TF binding motifs used in the study with their JASPAR accession numbers.

**Additional file 4:** Table S3. List of human bulk DNase-seq samples in BDDB.

**Additional file 5:** Table S4. List of mouse bulk DNase-seq samples in BDDB.

**Additional file 6:** Supplementary note. Choice of preprocessing parameters for identifying signal bins.

**Additional file 7:** Supplementary note. Derivation of moment estimators for $m_i$ and $s_i$.

**Additional file 8:** Review history.

---

**Peer review information**

**Acknowledgements**
We would like to thank the reviewers for their time and valuable feedback.

**Review history**
The review history is available as Additional file 8.

**Authors' contributions**
HJ conceived the study. HJ and ZJ developed the methods. ZJ implemented the methods and conducted data analysis. WZ processed scATAC-seq and bulk ATAC-seq data. ZJ and WH developed the software. All authors participated in writing the manuscript. All authors read and approved the final manuscript.

**Funding**
This study is supported by the National Institutes of Health grant R01HG010889.

**Availability of data and materials**
SCATE is freely available as an open source R package under the MIT License. One can download the software and source codes from the GitHub (https://github.com/zji90/SCATE) [40]. The version of source code used in this article is deposited in Zenodo with the access code DOI: 10.5281/zenodo.3711558 (https://doi.org/10.5281/zenodo.3711558) [41]. The SCATE R package is also submitted to Bioconductor [42] and is currently under review by the Bioconductor maintenance team. Once approved, it will be available at Bioconductor as well. Bulk DNase-seq data used to construct BDDB are downloaded from the ENCODE project [43]. Single-cell ATAC-seq data are downloaded from the Gene Expression Omnibus under the accession numbers GSE65360 [44], GSE96769 [45], GSE111586 [46], and GSE129785 [47]. Bulk ATAC-seq data for human hematopoietic and PBMC cell types are downloaded from the Gene Expression Omnibus under the accession number GSE74912 [48].

**Ethics approval and consent to participate**
Not applicable

**Consent for publication**
Not applicable

**Competing interests**
The authors declare that they have no competing interests.

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

## 
