## [**Additional file 8** Review history. · Genome Biology]

Review History

First round of review

Reviewer 1

Were you able to assess all statistics in the manuscript, including the appropriateness of statistical tests used?

Yes. Please refer to my comments below.

Were you able to directly test the methods? No.

Comments to author:

Ji et al. propose SCATE, a method that integrates three types of information: co-activated cis-regulatory elements (CREs), similar cells, and publicly available regulomic data to reconstruct activities of individual CREs by scATAC-seq data. My comments are below:

- 1) Most the empirical datasets are from the Buenrostro Lab: the GM12878 and K562 from Buenrostro et al., Nature 2015 and the hematopoietic differentiation data from Buenrostro et al., Cell 2018 and Corces et al., Nature Genetics 2016. These data were generated by isolating each individual cell and library was made in each cell chamber using machine such as Fluidigm. Compared to the more recently combinatorial indexing and the 10X Genomics data, this has much higher coverage and thus less sparsity, at the cost of throughput (i.e., the number of cells that can be sequenced). The only data that were not generated through Fluidigm was Cusanovich et al., Cell 2018, yet the authors demonstrated the method on a very easy case where mouse brain and thymus cells were mixed. These two clusters were highly discretized and the authors need to demonstrate SCATE on a more general split-and-pool or 10X Genomics platform.
- 2) For Figure 11G, benchmark analysis against existing scATAC-seq methods are needed, as there's quite variability in terms of performance within the category of "Binary" based on the recently benchmark paper Chen et al., Genome Biology 2019.
- 3) The first step of SCATE is to group co-activated CREs into clusters based on a pre-compiled list of candidate CREs with an additional option of user input. Does this list need to be cell-type specific? That is, would different cell types will have different co-activation patterns of CREs? In almost all analysis, the authors separate GM12878 from K562 (and mouse thymus from brain etc). However, in real analysis, one would not know the true underlying labels. In Figure 4, what would the pattern be like if I combine GM12878 and K562 together since I don't know the true labels but only have a pool of cells? Will this bias the estimated and if so, are there any solutions?
- 4) In addition to my previous concern, there is a potential circularity issue: SCATE takes as input a pre-defined list of CRE clusters, and through the model-based framework infers cluster-specific CRE activities. How much is simply transferred from the existing data, which can serve as a strong prior?
- 5) In Figure 3, correlations need to be included in a similar fashion to Figure 1F and Figure 1G. It seems pooling cells (of the same cell types) would significantly increase the correlation - is the cell-specific function $h_j()$ estimated from a group of cells?
- 6) The hierarchical model shown on page 16 should be moved to the sections where the model assumptions are first introduced. This helps me to piece together the different parts.
- 7) The method has been focused on CREs, while it has been shown that distal elements can also be informative for cell-type clustering. While the authors adopt a post-processing step, since SCATE still combines CREs, some more discussion on this would be helpful.

Reviewer 2

Were you able to assess all statistics in the manuscript, including the appropriateness of statistical tests used?

Yes.

Were you able to directly test the methods? Yes.

Comments to author:

The authors provide a set of novel methods to bridge the chromatin accessibility assays at single cell and bulk level. The experiments they designed to validate the sanity of their results are, to the most part, comprehensive. I particularly, liked the analysis about comparing the TF ChIP-seq peak data with the CRE activity obtained from SCATE. Comparing their results with the compatible state-of-the-art methods, showed a consistent superiority over those methods. However, there are several points that should be addressed by the authors.

Major:

- In lines 33 and 34 of page 4 the authors wrote: "Some CREs tend to have higher activity levels than others regardless of cell type (Fig. 1E: compare two CREs in blue boxes)". The authors need to provide more evidence that this is the case (having an example shown in Fig1E does not suffice for such a strong claim). It would be very helpful if they could use an integrative score capturing the variation of CRE activity across various cell types.
- Regarding the results related to Figure 5: The results obtained from the sc data of other methods don't necessarily have to look like the bulk signal, as the whole point of doing sc-seq is to have sub-samples of cells that are all mixed in the bulk-seq. So, the fact that some signal is missing in the single cell and not in bulk, doesn't mean the method didn't perform well.
- The statement provided in line 14 of page 9: how is this statistical theory applied to single cell data?
- In the first line of page 15, can the authors provide any justification why these numbers are chosen?
- In lines 27 and 28 of page 18, it states that the variable $\sim m_i$ is split into ten groups. But this variable is an estimate of log mean in CRE i , how can it be split into groups?
- Regarding the definition of function $h(\cdot)$ as described in page 18, can the authors further validate their selection of $h(\cdot)$ by using bulk DNase-seq and bulk ATAC-seq. Mainly to check if the properties assumed for function h hold (being monotone and also belonging to the class of spline functions).
- In the Stage 1 of SCATE's clustering approach described in page 19, can the authors provide what value of K they used. According to the statement in line 44 of this page "Each cluster contains approximately 100 CREs, ...", does it mean that they used 50 clusters? If so, what is the justification for choosing this number? And how does it change if the number is changed?
- I'm curious if the authors accounted for the cell cycle variations among the single cells. I didn't find any text related to that. However, this is an interesting point to be addressed when cell clustering is considered.

Minor:

- In lines 55 and 56 of page 7, the authors wrote: "Increasing cluster number implies decreasing cluster size." Is there a typo? Could it be that by the first occurrence of "cluster", they meant "cell"?
- In section "Analysis of a homogeneous cell population - a demonstration" and also in the subsequent text, the authors use the term homogeneous. How is this cell homogeneity defined?

Software:

-I was able to install and run the software. It is great that that the authors openly share their code under the MIT license. I was wondering whether it is possible to load peak calls instead of bam files? For the example data, it would be good to have the code for that as markdown in github, because copying from the pdf was quite cumbersome. Also it would be great if SCATE could be added to Bioconductor to increase its visibility.

Dear Editor,

Enclosed please find our revised manuscript “Single-cell ATAC-seq Signal Extraction and Enhancement with SCATE”. We would like to thank you and reviewers for your thoughtful comments which have greatly helped us to improve our manuscript. We have carefully considered and addressed all reviewers’ questions and have revised our manuscript accordingly. In the revised manuscript, changes are highlighted using the red color. Below is our point-by-point response to reviewers’ questions.

BLUE: Reviewer’s Questions and Comments

BLACK: Responses

RED: Revised texts in the article

Editor’s comment:

1. We will require that the source code is made publicly available under an open source license compliant with Open Source Initiative, with the license clearly stated in the manuscript.

[Response]: The source code of SCATE is made publicly available under the MIT License. This is stated in the Availability of Data and Materials section of the manuscript. The license statement can be found at <https://github.com/zji90/SCATE/blob/master/LICENSE>.

2. The source code should be deposited in a public repository, such as for instance github, with the accession links included in the manuscript. We also ask that the version of source code used in the manuscript is deposited in a DOI-assigning repository, such as zenodo, with the link also included.

[Response]: We have already made the source code publicly available on Github (<https://github.com/zji90/SCATE>) in our initial submission. In this revised submission, the version of source code used in the manuscript has also been deposited in Zenodo with DOI: 10.5281/zenodo.3711558. See the package <https://zenodo.org/record/3711558#.Xm6OIC2ZPRY>. The link is provided in the Availability of Data and Materials section of the manuscript.

3. All this information should be listed in a separate Availability of Data and Materials section of the manuscript.

[Response]: The above information is now listed in the Availability of Data and Materials section.

Reviewers' comments

Reviewer #1: ===

Ji et al. propose SCATE, a method that integrates three types of information: co-activated cis-regulatory elements (CREs), similar cells, and publicly available regulomic data to reconstruct activities of individual CREs by scATAC-seq data.

[Response]: We would like to thank the reviewer for the tremendously useful feedback. It has greatly helped us to improve our manuscript.

My comments are below:

1) Most the empirical datasets are from the Buenrostro Lab: the GM12878 and K562 from Buenrostro et al., Nature 2015 and the hematopoietic differentiation data from Buenrostro et al., Cell 2018 and Corces et al., Nature Genetics 2016. These data were generated by isolating each individual cell and library was made in each cell chamber using machine such as Fluidigm. Compared to the more recently combinatorial indexing and the 10X Genomics data, this has much higher coverage and thus less sparsity, at the cost of throughput (i.e., the number of cells that can be sequenced). The only data that were not generated through Fluidigm was Cusanovich et al., Cell 2018, yet the authors demonstrated the method on a very easy case where mouse brain and thymus cells were mixed. These two clusters were highly discretized and the authors need to demonstrate SCATE on a more general split-and-pool or 10X Genomics platform.

[Response]: This is a good question. We have now added analyses of a 10x Genomics scATAC-seq dataset on human peripheral blood mononuclear cells (PBMC). The results are presented in the new section titled "**Example 2: Analysis of 10x Genomics scATAC-seq data from human peripheral blood mononuclear cells (PBMC)**" and new **Figure 12** and **Additional file 2: Figure S12**. In this dataset, different cell types have a more complex clustering structure. Our results show that SCATE still outperforms the other methods for data generated using a platform such as 10X Genomics which has higher throughput than Fluidigm.

2) For Figure 11G, benchmark analysis against existing scATAC-seq methods are needed, as there's quite variability in terms of performance within the category of "Binary" based on the recently benchmark paper Chen et al., Genome Biology 2019.

[Response]: As suggested by the reviewer, we have now shown the performance of each individual method using their default parameters. For raw reads methods, we have added Dr.seq2 and scABC. For binary methods, we have added Cicero, Scasat, cisTopic and Destin. Another method PRISM uses binary accessibility as an intermediate step to compute cell distances. However, it does not export the accessibility matrix, nor did we find a way to modify its code to export this matrix. Therefore, PRISM is not compared. The comparison results between SCATE and these methods are added to **Figure 5, Additional file 2: Figure S4-Figure S6, Figure S8-Figure S9, Figure S11-Figure S12**, and they are discussed in relevant places in the manuscript

using red colored texts. These analyses show that SCATE robustly outperformed these existing methods.

3) The first step of SCATE is to group co-activated CREs into clusters based on a pre-compiled list of candidate CREs with an additional option of user input. Does this list need to be cell-type specific? That is, would different cell types will have different co-activation patterns of CREs? In almost all analysis, the authors separate GM12878 from K562 (and mouse thymus from brain etc). However, in real analysis, one would not know the true underlying labels. In Figure 4, what would the pattern be like if I combine GM12878 and K562 together since I don't know the true labels but only have a pool of cells? Will this bias the estimated and if so, are there any solutions?

[Response]: These are very good questions. Below we address them one-by-one.

3.1) Does the CRE list need to be cell type specific?

This is a great question. By default, SCATE will use its precompiled CRE list and CRE clusters as input. These precompiled list and clusters are constructed using the bulk DNase-seq database BDDDB. For users who choose to run SCATE under its default setting and for a given version of BDDDB, these precompiled CRE list and clusters will be fixed and remain the same for all new scATAC-seq datasets. The reviewer raised a very good point which we are also fully aware of, that is, a new scATAC-seq dataset may contain new CREs and new CRE correlation structures that may not be fully captured by our precompiled CRE list and clusters. For this reason, we provided functions in SCATE that allow users to compile dataset-specific CRE list and clusters for their own scATAC-seq data. Despite this, we choose to use the BDDDB precompiled CRE list and clusters as the default mode because it is convenient for users to use and generally produces very competitive performance. To explain this in detail, we compared three different ways to run SCATE:

- Approach 1 (Default: SCATE – BDDDB): This is the default SCATE that uses the precompiled CRE list and CRE clusters constructed from BDDDB.
- Approach 2 (SCATE – User Data): Here the input CRE list and CRE clusters are compiled from user's own scATAC-seq data. No BDDDB information is used. Suppose users want to analyze a scATAC-seq dataset, we will first cluster cells using the scATAC-seq data. For each cell cluster, cells are pooled to create a pseudo-bulk sample. Then CREs in each pseudo-bulk sample are identified and the union of CREs from all pseudo-bulk samples (i.e. all cell clusters) will generate the input CRE list. These CREs are clustered based on their co-activation patterns in the scATAC-seq data. The resulting CRE list and CRE clusters reflect the cell types in the scATAC-seq data that users want to analyze. They will be used as the input for SCATE. They do not use any information in BDDDB.
- Approach 3 (SCATE – BDDDB + User Data): Here users augment BDDDB using their scATAC-seq data. The CREs obtained from Approach 1 (from BDDDB) and Approach 2 (from the scATAC-seq data users want to analyze) are combined. Their union are used as the input CRE list. The pseudobulk samples obtained from Approach 2 are added to

BDDDB to expand BDDDB. CREs are clustered using the expanded BDDDB. In this approach, the CRE list and CRE clusters use information from both BDDDB and users' own scATAC-seq data. They will be used as the input for SCATE.

Based on our experience with real data, we found that

- (1) SCATE(BDDDB) usually performs better than SCATE(User Data) (see the new **Additional file 2: Fig. S13**). This is likely because CREs and their clustering patterns compiled from diverse cell types in BDDDB are more informative than those compiled from a limited number of cell types in a new scATAC-seq data.
- (2) SCATE(BDDDB) and SCATE(BDDDB+User Data) usually show similar performance, with SCATE(BDDDB+User Data) being slightly better (**Additional file 2: Fig. S13**). Despite the slight loss of accuracy compared to SCATE(BDDDB+User Data), SCATE(BDDDB) is substantially easier to use. In order to use SCATE(BDDDB+User Data), users have to download the DNase-seq data in BDDDB and run CRE detection and clustering themselves which require extensive computation (it typically takes 1-2 days in a computer with 20 cores (2.5 GHz CPU/core)). By contrast, in order to use SCATE(BDDDB), one can skip these tedious and computation heavy steps and only download the precompiled CRE list and clustering. With these precompiled CREs and CRE clusters, running SCATE only takes a few minutes per cell cluster.

Note that in all our benchmark analyses, the cell types that appeared in the scATAC-seq data were removed from the BDDDB so that CREs constructed using BDDDB did not use cell types involved in the scATAC-seq data. In this way, the evaluation performance we obtained can reliably reflect the method's performance when applying SCATE to new datasets which involve new cell types. Also note that our saturation analysis in **Additional file 2: Figure S1** shows that most of the CREs one would see in a new dataset typically are already covered by the BDDDB precompiled CRE list.

Taking all these into account, we chose to use the BDDDB precompiled CRE list and CRE clusters (i.e. SCATE(BDDDB)) as the default mode of SCATE in order to balance the performance and easiness to use the software. This is now discussed in the Discussion section and **Additional file 2: Fig. S13**. Please refer to the paragraphs in the **Discussion** section starting with

“A potential limitation of using our precompiled CRE list and clusters is that for a given version of BDDDB, these lists will be fixed and remain the same for analyzing all new scATAC-seq datasets. A new scATAC-seq dataset may contain new CREs and new CRE correlation structures that may not be fully captured by our precompiled CRE list and clusters. ...”

3.2) In almost all analysis, the authors separate GM12878 from K562 (and mouse thymus from brain etc). However, in real analysis, one would not know the true underlying labels.

The reviewer is absolutely right that in real analysis, one may not know the true underlying labels. This is exactly why in addition to analyzing GM12878 and K562 separately in the sections titled “Analysis of a homogeneous cell population ...”, we also created mixtures of GM12878 and K562 cells and analyzed these cell mixtures in the section titled **“Analysis of a heterogeneous cell population - demonstration and systematic evaluation”**. When analyzing these cell mixtures, we intentionally pretended that cells' true cell type labels are unknown and did not

use these labels for running SCATE and other methods. This is to mimic how data would be analyzed in reality. Cells' true labels are only used for evaluating the analysis results after the scATAC-seq analysis is done (we need cell type labels to match cell clusters with their corresponding bulk gold standard regulome data). Our results in **Figure 9 and Additional file 2: Figure S8** show that SCATE outperformed the other methods for analyzing these mixtures.

Moreover, in order to further demonstrate and evaluate SCATE in a more realistic setting, we have analyzed human hematopoietic differentiation scATAC-seq data in section "**Example 1: Analysis of scATAC-seq data from human hematopoietic differentiation**". We also added a new analysis of human PBMC 10x Genomics scATAC-seq data in section "**Example 2: Analysis of 10x Genomics scATAC-seq data from human peripheral blood mononuclear cells (PBMC)**". These datasets are more complex and reflect the reality that (1) the data is a heterogeneous population of cells from multiple cell types, and (2) unsupervised cell clustering may not perfectly separate cells originating from different cell types. The analyses in these realistic applications also show that SCATE outperformed the other methods.

We have now revised manuscript by adding texts in various places to emphasize the fact that:

"In all these analyses, we pretended that cells' true cell type labels were unknown and did not use them." (in section "**Analysis of a heterogeneous cell population - demonstration and systematic evaluation**")

"However, they were not used in our SCATE analyses so that our results reflect how data would be analyzed in reality. The true cell type labels were only used after the analysis to evaluate methods." (in section "**Example 1: Analysis of scATAC-seq data from human hematopoietic differentiation**")

"Again, in order to illustrate how scATAC-seq data would be analyzed in reality, we pretended that cells' true cell type labels are unknown when running SCATE and other methods." (in section "**Example 2: Analysis of 10x Genomics scATAC-seq data from human peripheral blood mononuclear cells (PBMC)**")

3.3) In **Figure 4**, what would the pattern be like if I combine **GM12878** and **K562** together since I don't know the true labels but only have a pool of cells? Will this bias the estimated and if so, are there any solutions?

This question is now answered in the first two paragraphs of the section "**Analysis of a heterogeneous cell population - demonstration and systematic evaluation**" and new **Additional file 2: Figure S7**:

"The analyses of homogeneous cell populations provide a demonstration of the basic building block of SCATE. In reality, however, scATAC-seq is usually used to analyze a heterogeneous cell population consisting of multiple cell types where the cell type labels are unknown. To analyze such a heterogeneous cell population, one usually will first computationally cluster cells into relatively homogeneous subpopulations and then analyze each cell cluster as a homogeneous population. Due to inevitable noises, each cell cluster obtained in this way may not be pure. For example, while the majority of cells in a cell cluster may be of one cell type, the cluster may also contain cells from other cell types. As data analysts do not know cells' true cell type labels, they can only treat all cells in the same cluster as if they were one cell type."

*“In order to see how SCATE tunes the analysis resolution when a cell cluster contains noise, we mixed K562 and GM12878 cells with different ratios (K562:GM12878 = 100%:0%, 80%:20%, 60%:40%) to mimic a cell cluster dominated by K562 cells but with different levels of noises introduced by GM12878 cells. The cross-validation procedure of SCATE was used to select the CRE cluster number as in Figure 4. The analysis was repeated by setting the total cell number to 10 and 100 respectively. **Additional file 2: Figure S7** shows that as the number of cells increased, the number of CRE clusters chosen by SCATE also increased regardless of the noise level. This indicates that when more reads are available by pooling more cells, SCATE will increase the analysis resolution. In most cases, the optimal CRE cluster number chosen by SCATE was largely consistent with the true optimal CRE cluster number determined by comparing scATAC-seq with bulk K562 DNase-seq. The only exception is when the noise level was high (K562:GM12878 = 60%:40% for cell number=100, where the optimal cluster number chosen by SCATE was bigger than the optimal cluster number based on bulk K562 DNase-seq). In that case, however, one can argue that K562 bulk DNase-seq data may not reflect the chromatin profile of a mixture of K562 and GM12878 cells, but SCATE attempts to optimize the signal reconstruction for a cell cluster which is a mixture of K562 and GM12878 cells with almost equal proportion. Therefore they try to measure different things and one should not expect that the optimal cluster number determined by K562 bulk DNase-seq will be consistent with the cluster number chosen by SCATE. This is different from Figure 4 where the cell type measured by bulk gold standard is consistent with the cells analyzed by SCATE.”*

To summarize, in reality a heterogeneous population is analyzed by first clustering cells into relatively homogeneous clusters and then analyzing each cell cluster as a homogeneous population. Each cell cluster may not be completely pure. For a cell cluster dominated by one cell type X but with a small to moderate proportion of contamination cells from other cell types, the optimal CRE cluster number K (which controls the analysis resolution) chosen by SCATE is still consistent with the optimal K one would obtain based on the bulk gold standard DNase-seq in cell type X. For a cell cluster where cell type X is contaminated by a large proportion of cells from other cell types, the optimal CRE cluster number K chosen by SCATE may not necessarily be consistent with the optimal K obtained using the bulk DNase-seq in cell type X. In this case, however, the interpretation of the perceived “bias” is tricky because it is unclear what the true gold standard should be for evaluating a non-pure cell cluster. For example, in the above analysis, K562 bulk DNase-seq is used as gold standard to evaluate how SCATE tunes analysis resolution in a cell cluster consisting of K562 cells contaminated by GM12878 cells. Since they measure different things (pure K562 vs. a mixture of K562 and GM), even if the optimal CRE cluster number chosen by bulk K562 DNase-seq is different from the optimal CRE cluster number chosen by SCATE, it does not mean that SCATE did anything wrong in terms of selecting the analysis resolution. SCATE only did what it is supposed to do: increasing analysis resolution when there is more data. In other words, the perceived “bias” is not due to the parameter tuning algorithm (because our Figure 4 shows that when cells in the cell cluster have a cell type consistent with the bulk gold standard DNase-seq, the tuning procedure worked well). If there would be any problem with the perceived “bias”, the problem usually comes from the earlier cell clustering stage where cells were not grouped into pure clusters (which can happen for many different reasons, e.g., when the data have high noise level, or when cell types are intrinsically difficult to separate, or when the cell clustering algorithm is not optimal, etc.). Thus, if you ask whether there is any solution to this problem, solutions perhaps are not in the procedure for tuning the analysis resolution but should be in earlier steps (i.e. obtaining better cell clustering by

generating cleaner scATAC-seq data or developing better cell clustering algorithms). Computationally, such solutions sometimes are possible (e.g., if the “bias” is due to poor cell clustering algorithm) and sometimes are impossible (e.g. if the “bias” is due to noisy data which has already been generated).

4) In addition to my previous concern, there is a potential circularity issue: SCATE takes as input a pre-defined list of CRE clusters, and through the model-based framework infers cluster-specific CRE activities. How much is simply transferred from the existing data, which can serve as a strong prior?

[Response]: We are sorry that our original description may not be clear enough, but as we will explain below, there is no circularity issue.

The precompiled list of CRE clusters contain information about which CREs are correlated. For example, if CRE X, CRE Y and CRE Z are in the same cluster, then they tend to be co-activated. In other words, if CRE X and CRE Y has high activity in cell type A and low activity in cell type B, then CRE Z often will also have high activity in cell type A and low activity in cell type B. This correlation is learned using a large number of DNase-seq samples representing diverse cell types in BDDDB. This correlation information does not tell you the actual activity level of CRE X, CRE Y and CRE Z in the new scATAC-seq data. To infer CREs' actual activity in scATAC-seq, one has to use information from the scATAC-seq (i.e., read counts at CRE X, CRE Y and CRE Z in the scATAC-seq data). If they all have high counts, then it suggests that CREs X, Y and Z are highly active in the new scATAC-seq data. If they all have low counts, then it suggests that CREs X, Y and Z have low activity in the new scATAC-seq data. Thus only the information about how CREs are correlated are transferred from the existing data to the new scATAC-seq data, and such information is not enough to determine the actual activity of CREs in the new scATAC-seq data. To infer CREs actual activities in scATAC-seq, one still need scATAC-seq read counts.

The transferred correlation information is helpful for improving the estimation of CRE activities. Following our example above, consider two scenarios: (1) CRE Z has 0 read, but CRE X and CRE Y both have 1 read; (2) CRE Z has 0 read, and CRE X and CRE Y also have 0 read. Since we know these three CREs are correlated based on the prior information transferred from the precompiled CRE clusters, one can infer that CRE Z is more likely to be active in scenario (1) than in scenario (2), because its correlated CREs X and Y are active in scenario (1) but not in scenario (2). In other words, the zero read count for CRE Z in scenario (1) is more likely to represent a noisy measurement, whereas the zero read count for CRE Z in scenario (2) more likely reflects its real low activity level. This example shows how the prior correlation information is combined with the new scATAC-seq data to help infer CRE activities.

Essentially, combining prior information from the precompiled CRE clusters can be viewed as a way to “regularize” the estimation of CRE activities from the observed scATAC-seq read count data. The essence of regularization in statistics and machine learning is bias-variance tradeoff. Inevitably, prior information will introduce bias in exchange for reduced variance and improved overall estimation performance. One question is whether the correlation structure learned from BDDDB is too strong and does not reflect CRE's correlation structure in the scATAC-seq data.

This is related to reviewer's question 3.1. However, as we discussed in our responses to question (3.1), our comparisons of SCATE(BDDB), SCATE(User Data) and SCATE(BDDB+User Data) in **Additional file 2: Fig. S13** show that empirically, using our precompiled CRE clusters (SCATE(BDDB)) actually performs better than using CRE clusters constructed using the scATAC-seq data itself (SCATE(User Data)). This is not completely unexpected because the precompiled CRE clusters used large amounts of information from diverse cell types not available in the scATAC-seq data. **Additional file 2: Fig. S13** also shows that using our precompiled CRE clusters (SCATE(BDDB)) also performed well compared to clustering CREs by combining the existing data in BDDB and new scATAC-seq data (SCATE(BDDB+User Data)). Despite the slight loss of accuracy compared to SCATE(BDDB+User Data), SCATE(BDDB) is much easier to use. Collectively, these data show that our precompiled CRE clusters helped transfer information from the existing data to scATAC-seq data at an appropriate level that balances the performance and easiness to use the software.

5) In Figure 3, correlations need to be included in a similar fashion to Figure 1F and Figure 1G. It seems pooling cells (of the same cell types) would significantly increase the correlation - is the cell-specific function $h_j()$ estimated from a group of cells?

[Response]: We have now displayed the correlation coefficients in Figure 3 similar to Figure 1F and 1G. Pooling cells of the same cell types did increase the correlation. The cell-specific function $h_j(.)$ is estimated from a group of cells (i.e., cells in each cell cluster are pooled to create a pseudobulk sample, and $h_j(.)$ for cell cluster j is estimated using its corresponding pseudobulk sample). This is now clarified in the manuscript **Methods section "SCATE for multiple cells"**:

"For each cell cluster, reads from all cells are pooled together to create a pseudo-cell. The SCATE model for a single cell described above is then applied to the pseudo-cell to estimate CRE activities. For instance, the bias normalizing function $h_j(.)$ is estimated by treating the pseudo-cell obtained from cluster j (after pooling cells) as a single cell."

6) The hierarchical model shown on page 16 should be moved to the sections where the model assumptions are first introduced. This helps me to piece together the different parts.

[Response]: This is a good suggestion. We have now moved the model to where the model assumptions are first introduced, both on page 4 and page 21 (see red colored texts in the manuscript).

7) The method has been focused on CREs, while it has been shown that distal elements can also be informative for cell-type clustering. While the authors adopt a post-processing step, since SCATE still combines CREs, some more discussion on this would be helpful.

[Response]: We note that the main novelty and primary goal of SCATE is to estimate the activities of individual CREs rather than clustering cells. This is different from many other tools whose

primary goal is to cluster cells into cell types. For cell clustering, the default clustering in SCATE is provided for user's convenience, but we also provide users with the option to use their own cell clustering results obtained from other tools. Although our default cell clustering algorithm does not treat distal and proximal CREs differently, users has the flexibility to replace the default SCATE cell clustering by cell clustering obtained from other tools that treat distal and proximal regulatory elements differently (e.g. *Destin*). SCATE will then reconstruct chromatin profiles for each of the user-provided cell cluster.

Given cell clustering, SCATE tries to estimate activity of each individual CRE in each cell cluster. For this step, we do not distinguish between proximal (e.g. promoter) and distal (e.g. enhancers) cis-regulatory elements. SCATE will estimate the activities for both types of CREs. Note that the input CREs for SCATE are compiled from DNase-seq or ATAC-seq data which cover both proximal and distal elements. When we use these data to detect CREs, proximal and distal elements were not treated differently (e.g., if there is a peak, it will be detected regardless of whether the peak is distal or proximal). We have now discussed this in the Discussion section:

“Since many methods for clustering cells using scATAC-seq data have been developed (Additional file 1: Table S1), cell clustering per se is not the focus of this article. In principle, the SCATE model may be coupled with any cell clustering method. ... As another example, some recent studies suggest that distal regulatory elements such as enhancers may be more informative for clustering cells compared to proximal elements such as promoters \cite{urrutia2018destin}. Although this information is not currently considered in our default cell clustering algorithm, users have the flexibility to replace the default SCATE cell clustering by cell clustering obtained from other tools (e.g., \cite{urrutia2018destin}) that treat distal and proximal regulatory elements differently. Once cell clustering is given, SCATE will apply the same algorithm to estimate activities of all CREs regardless of whether they are proximal or distal. We note that the input CREs for SCATE are compiled from DNase-seq or ATAC-seq data which cover both proximal and distal elements. When we use these data to detect CREs, proximal and distal elements were not treated differently.”

Reviewer #2:

The authors provide a set of novel methods to bridge the chromatin accessibility assays at single cell and bulk level. The experiments they designed to validate the sanity of their results are, to the most part, comprehensive. I particularly, liked the analysis about comparing the TF ChIP-seq peak data with the CRE activity obtained from SCATE. Comparing their results with the compatible state-of-the-art methods, showed a consistent superiority over those methods. However, there are several points that should be addressed by the authors.

[Response]: We would like to thank the reviewer for providing a nice summary. Your feedback is tremendously useful. It has greatly helped us to improve our manuscript.

Major:

1) - In lines 33 and 34 of page 4 the authors wrote: "Some CREs tend to have higher activity levels than others regardless of cell type (Fig. 1E: compare two CREs in blue boxes)". The authors need to provide more evidence that this is the case (having an example shown in Fig1E does not suffice for such a strong claim). It would be very helpful if they could use an integrative score capturing the variation of CRE activity across various cell types.

[Response]: This is a great suggestion. We have now added quantitative evidence and a new **Additional file 2: Figure S2** to support our claim. We use the average DNase-seq profile across all BDDB samples to characterize the average activity level of each CRE. This mean DNase-seq profile reflects locus-specific CRE activity. We asked how this mean DNase-seq profile correlated with the DNase-seq profile of each individual cell type (i.e., how well the locus-specific but cell-type-independent mean CRE activity can explain CRE activity in each individual cell type). Our findings are summarized in the revised manuscript (page 4) as follows:

"In fact, 55.7% of the total data variance in BDDB human DNase-seq samples is explained by the mean human DNase-seq profile, and 60.1% of the total data variance in BDDB mouse DNase-seq samples is explained by the mean mouse DNase-seq profile (Methods). The Pearson correlation coefficient between the mean DNase-seq profile and each individual DNase-seq sample in the BDDB is bigger than 0.5 for most of the samples, and the median correlation is 0.78 for human and 0.81 for mouse (Additional file 2: Fig. S2)."

This means that simply using the locus-specific but cell-type-invariant mean CRE activity (i.e. mean DNase-seq profile), one can explain a large proportion of the variation in each individual cell type.

2) - Regarding the results related to Figure 5: The results obtained from the sc data of other methods don't necessarily have to look like the bulk signal, as the whole point of doing sc-seq is to have sub-samples of cells that are all mixed in the bulk-seq. So, the fact that some signal is missing in the single cell and not in bulk, doesn't mean the method didn't perform well.

[Response]: We agree with the reviewer that data in a single cell do not necessarily need to look like the bulk signal due to the cell heterogeneity. However, this does not mean that bulk signal does not provide information for comparing different scATAC-seq analysis methods. There are four important points that need to be considered.

First, in Figure 5 we not only compared single cell with bulk signal, but we also pooled single cells into pseudobulk samples and compared pseudobulk samples with bulk signal. The pseudobulk samples represent the average of many cells. As more and more cells are pooled together, the cell heterogeneity and stochasticity are averaged out. As a result, the pseudobulk samples should look more and more similar to the bulk signal. Figure 5 shows that for pseudobulk samples created by pooling multiple cells (25 or 100 cells), SCATE still outperformed the other methods for capturing the true bulk signal.

Second, if one believes that the goal of scATAC-seq is to allow study of cell heterogeneity, then a good method should have the ability to accurately capture cell differences. Importantly, difference between two cell types is a special case of cell heterogeneity. A good method should be able to keep cell type differences when comparing two cells or two pseudobulk samples from two different cell types. If a method cannot reliably capture the differential signals between two cell types, it is hard to argue that the method can provide a reliable approach to study cell heterogeneity. Figure 5 shows that SCATE best captured both the cell-type-specific signals when comparing K562 and GM12878 and signals that are shared by the two cell types. In this regard, SCATE again outperformed the other methods.

Third, Figure 5 is only one specific example to illustrate the comparison between SCATE and the other methods. The most systematic comparisons are conducted in Figures 6-9. In these comparisons, the analyses were done by randomly sampling different cells and pooling different numbers of cells. In all these comparisons, SCATE robustly performed among the best. For example, if a method can better keep cell heterogeneity, it should be able to better keep differences between two cell types. In repeated samplings of cells, the differential scATAC-seq signals reconstructed by a better method therefore should show better overall correlation with the bulk differential signal. This is observed for SCATE in Figure 9 F,I,L,O but not for the other methods.

Fourth, ideally one would like to evaluate single-cell analysis methods using gold standard signals at the single-cell level. However, single-cell resolution gold standard is difficult to obtain. Thus, evaluation of different methods has to primarily rely on bulk signals. Fortunately, because of the three points above, bulk signal still provides valuable information to assess the relative performance of different methods.

We hope that this discussion will clarify the relevance of our Figure 5 and other evaluations based on bulk signals. We have now added these discussions in various places of the manuscript:

(Page 7): *“For method evaluation, ideally one would like to have a gold standard for each single cell. However, single-cell resolution gold standard is difficult to obtain. For this reason, our method evaluation primarily relied on comparing scATAC-seq signals reconstructed from a single-cell or by pooling multiple cells to bulk DNase-seq or ATAC-seq signals. In this regard, one may view single cells as random samples from a cell population, and bulk signal characterizes cells’ mean behavior in the cell population. Although each cell is not exactly the same as the population mean, its behavior should fluctuate around the mean. Moreover, one should expect that the pseudobulk signal obtained by pooling an increasing number of cells should become increasingly more similar to the true bulk signal.”*

(Page 10): *“While it is also possible that signals in a single cell do not necessarily need to look like the bulk signal due to cell heterogeneity and hence explaining why signals generated by Raw reads, Binary and CRE cluster methods in a single cell were different from the bulk signal, Figure 5 shows that SCATE also outperformed these methods when pooling multiple cells into pseudobulk samples (e.g., pooling 25 and 100 cells), suggesting that the better performance of SCATE is real.”*

(Page 14): *“Note that if one views scATAC-seq as a tool for studying cell heterogeneity, then a good analysis method should have the ability to accurately capture differences among cells. Importantly, since differences between cell types are a special case of cell heterogeneity, a good*

method should be able to keep cell type differences when comparing two cells or two pseudo-bulk samples from two different cell types.”

3) - The statement provided in line 14 of page 9: how is this statistical theory applied to single cell data?

[Response]: We are sorry that our original language may not have accurately conveyed what we want to say. What we wanted to say was that each cell may be viewed as a realization of a random vector X . The theoretical mean of X , denoted as $E[X]$, characterizes the average behavior of the cell population (it corresponds to the bulk signal). In statistics, when $E[X]$ is unobserved, one can use the observed X from multiple single cells to compute a sample mean \bar{X} . The sample mean \bar{X} provides an unbiased estimator for $E[X]$. Thus the pooled single cell signals (\bar{X}) and bulk signals ($E[X]$) should be comparable to each other. Since this involves too many technical languages, we have now rephrased it in this revised manuscript as follows:

(Page 7): *“For method evaluation, ideally one would like to have a gold standard for each single cell. However, single-cell resolution gold standard is difficult to obtain. For this reason, our method evaluation primarily relied on comparing scATAC-seq signals reconstructed from a single-cell or by pooling multiple cells to bulk DNase-seq or ATAC-seq signals. In this regard, one may view single cells as random samples from a cell population, and bulk signal characterizes cells’ mean behavior in the cell population. Although each cell is not exactly the same as the population mean, its behavior should fluctuate around the mean. Moreover, one should expect that the pseudobulk signal obtained by pooling an increasing number of cells should become increasingly more similar to the true bulk signal.”*

As stated in our response to your last question, we acknowledge the limitation of using bulk signal for evaluation (because a single cell do not necessarily need to look like the bulk signal due to the cell heterogeneity). However, due to the lack of single-cell level gold standard, there are not many options available for evaluating single-cell methods other than using bulk signals as gold standard. Importantly, in our response to your last question, we also explained in detail why bulk signal, despite its potential limitation, can still provide valuable information for evaluating single cell methods.

4) - In the first line of page 15, can the authors provide any justification why these numbers are chosen?

[Response]: We have now explained the rationale in the revised manuscript, the new **Additional file 6: Supplementary Note** and **Additional file 2: Figure S14**.

(Page 20): *“The cutoffs for defining signal bins are used to filter out noisy genomic loci since including such loci will increase computational burden. For example, the CRE clustering below failed to run on our computer when we included all genomic bins in the analysis. We explored different choices of cutoffs that were computationally feasible on our computer and found that*

the cutoffs used above had good empirical performance compared to using looser or more stringent cutoffs (see details in Additional file 2: Fig. S14 and Additional file 6: Supplementary Note)."

Please see **Supplementary Note** and **Additional file 2: Figure S14** for details.

5) - In lines 27 and 28 of page 18, it states that the variable \tilde{m}_i is split into ten groups. But this variable is an estimate of log mean in CRE i , how can it be split into groups?

[Response]: We are sorry that our original description was not clear enough. Each CRE has a m_i , but there are many CREs. We collected m_i s from all CREs and computed their quantiles. One can group CREs into 10 strata based on their m_i values. For example, CREs with the smallest m_i values (i.e. $m_i < 10\%$ quantile) are in one stratum, CREs with m_i values between 10% and 20% quantiles belong to another stratum, etc. In this way, m_i s from all CREs are grouped into 10 strata. This is equivalent to say that CREs are grouped into ten strata based on their mean activity levels. Within each stratum, we then selected CREs with the lowest variability by finding 1000 CREs with the smallest s_i s in that stratum. This is now clarified in the manuscript as follows:

(Page 23): "*To select low-variability CREs, we first group all known CREs into 10 strata based on their baseline mean activity values (i.e. \tilde{m}_i s). To do so, the \tilde{m}_i s from all CREs are collected and their 10%, 20%, ..., 90% quantiles are computed. These quantiles are used to define the 10 strata.*"

6) - Regarding the definition of function $h(\cdot)$ as described in page 18, can the authors further validate their selection of $h(\cdot)$ by using bulk DNase-seq and bulk ATAC-seq. Mainly to check if the properties assumed for function h hold (being monotone and also belonging to the class of spline functions).

[Response]: We have now added **Additional file 2: Figure S3** that shows $h(\cdot)$ (on exponential scale) fitted to normalizing bulk ATAC-seq data and bulk DNase-seq data in BDDB. The analysis shows that the monotone splines we used were able to capture the patterns observed in bulk data too.

Note that both in **Figure 3** and **Additional file 2: Figure S3**, y-axis shows the observed read counts rather than log counts (since the single cell data have too many zero counts). Accordingly, $\exp\{h(x)\}$ instead of $h(x)$ is shown in order to be consistent with the scale (note that $h(x)$ models log activity). These plots show that assuming $\exp\{h(x)\}$ is a monotone function of x is reasonable. This also implies that $h(x)$ is a monotone function of x . In other words, if $\exp\{h(x)\}$ is monotone, then increasing x will increase $\exp\{h(x)\}$, which also implies that $h(x)$ will increase since $\exp\{y\}$ is a monotone function of y .

7) - In the Stage 1 of SCATE's clustering approach described in page 19, can the authors provide what value of K they used. According to the statement in line 44 of this page "Each cluster contains approximately 100 CREs, ...", does it mean that they used 50 clusters? If so, what is the justification for choosing this number? And how does it change if the number is changed?

[Response]: We are sorry that our original description was not clear enough which obviously caused the reviewer to misunderstand our procedure.

In Stage 1, K-means clustering (Euclidean distance) is used to group all CREs into 5000 clusters. The end result of this stage is 5000 clusters, and each cluster consists of a group of co-activated CREs. The mean number of CREs contained in each cluster is approximately 100 (human: 522,173 CREs/5000 clusters = 104 CREs/cluster; for mouse: 475,865 CREs/5000 clusters = 95 CREs/cluster). We report this number to give readers a sense of how big a cluster typically is. It does not mean that we further divide CREs within each cluster into smaller clusters at this stage. To make it clear, there is no further clustering at stage 1. Thus the questions "does it mean that they used 50 clusters" and "what is the justification for choosing this number" and "how does it change if the number is changed" are irrelevant.

In Stage 2, we grouped the 5000 clusters obtained from Stage 1 to obtain coarser clusters. To do so, the 5000 clusters from stage 1 are grouped hierarchically using hierarchical clustering (Euclidean distance, complete agglomeration) based on their mean activity profile. In this way, CREs are hierarchically grouped into 5000, 2500, 1250, 625, 312 and 156 clusters.

In Stage 3, we divided each of the 5000 clusters obtained from stage 1 into finer clusters. To do so, for each cluster obtained in Stage 1, hierarchical clustering is applied to split CREs in that cluster into smaller clusters. Conceptually, each cluster from Stage 1 can be divided into 2, 4, 8, ... subclusters until each subcluster contains only 1-2 CREs. For different Stage 1 clusters, their CRE numbers are different and therefore the exact number of their subclusters may vary.

After all three stages, we obtain clusters of CREs at multiple granularity levels. In other words, CREs are grouped into K clusters for different K values. For human, K = 156, 312, 625, 1250, 2500, 5000, 9856, 19008, 35361, 64398, 117596, 213432, 521820. For mouse, K = 156, 312, 625, 1250, 2500, 5000, 9996, 19953, 39732, 78868, 154813, 283422, 465055.

The optimal cluster number used to fit SCATE model is then chosen from these Ks using the cross-validation procedure described in our manuscript.

We have now clarified these in the revised manuscript. Please see the red highlighted texts on page 25 of the manuscript.

8) - I'm curious if the authors accounted for the cell cycle variations among the single cells. I didn't find any text related to that. However, this is an interesting point to be addressed when cell clustering is considered.

[Response]: This is a good question. Cell cycle variation is an open problem we are exploring now. However, we currently do not have a good solution for scATAC-seq data. Addressing this problem requires robust methods to accurately infer cells' phase in cell cycle using scATAC-seq data and systematic benchmark datasets and method evaluation. Both are non-trivial for

scATAC-seq and are beyond the scope of this study. For cell clustering, if users have their own clustering that accounted for cell cycle, they could replace the default SCATE clustering using their own clustering. This provides users with some flexibility to deal with this issue if they have their own solution to this problem. We have now discussed this in the Discussion section:

“Since many methods for clustering cells using scATAC-seq data have been developed (Additional file 1: Table S1), cell clustering per se is not the focus of this article. In principle, the SCATE model may be coupled with any cell clustering method. While our implementation uses model-based clustering as the default, users are provided with the option to use their own cell clustering results as the input for SCATE. For example, cell clustering may be influenced by cell cycle which is not adjusted for in the default clustering method in SCATE. However, if users want to adjust for cell cycle and have performed their own cell clustering to do so, they could replace the default SCATE clustering with their own cell clustering.”

“In the future, the SCATE framework may be extended in multiple directions. For example, how should one account for the effects of cell cycles in scATAC-seq analysis remains an open problem. Addressing this problem requires robust methods to accurately infer cells' phase in cell cycle using scATAC-seq data and systematic benchmark datasets and method evaluation. Both are non-trivial for scATAC-seq and are beyond the scope of this study. However, they are interesting topics for future research.”

Minor:

- In lines 55 and 56 of page 7, the authors wrote: "Increasing cluster number implies decreasing cluster size." Is there a typo? Could it be that by the first occurrence of "cluster", they meant "cell"?

[Response]: This is not a typo but we probably did not phrase it clearly. What we intended to say is that increasing the number of CRE clusters (i.e. increasing cluster number) implies that the average number of CREs in each CRE cluster will decrease (i.e. decreasing cluster size). This is because the total number of CREs is fixed. If you group them into more clusters, each cluster will contain fewer CREs. We now reworded this sentence as follows to make it clearer:

(Page 9): *“Increasing the number of CRE clusters implies that the average number of CREs in each CRE cluster will decrease because the total number of input CREs is fixed.”*

- In section "Analysis of a homogeneous cell population - a demonstration" and also in the subsequent text, the authors use the term homogeneous. How is this cell homogeneity defined?

[Response]: This is now explained in the manuscript as follows:

(Page 7): *“It should be pointed out that “homogeneous” is a relative concept rather than an absolute one since one can always define cell subtypes in a cell population by computationally grouping cells into clusters and subclusters at different granularity levels. In this study, “homogeneous” is technically defined as the finest granularity level for which we were able to obtain the corresponding bulk gold standard regulome data for method evaluation. We use this tech-*

nical definition for two reasons. First, even if a test cell type may potentially be decomposed further into multiple cell subtypes, we could not conduct the benchmark analysis at the cell subtype level if the gold standard bulk regulome data for those cell subtypes are unavailable and the true subtype label of each cell is unknown. Second, the primary goal of our analysis of a homogeneous cell population is to serve as a bridge to help readers understand how SCATE would analyze each cell cluster in a heterogeneous cell population. Our working definition of "homogeneous" is sufficient to meet this need."

As an example, for K562 scATAC-seq data, we have K562 bulk DNase-seq data as gold standard, but we don't have gold standard for cell subpopulations within K562, thus we view K562 as a homogeneous cell population for evaluating methods. For synthetic data with a mixture of K562 and GM12878 cells, we have bulk DNase-seq data for both cell subpopulations (K562 and GM12878), thus the data are viewed as a heterogeneous population for evaluating methods.

Software:

-I was able to install and run the software. It is great that that the authors openly share their code under the MIT license. I was wondering whether it is possible to load peak calls instead of bam files? For the example data, it would be good to have the code for that as markdown in github, because copying from the pdf was quite cumbersome. Also it would be great if SCATE could be added to Bioconductor to increase its visibility.

[Response]: These are good suggestions.

For the input files, we have now added support to load peak calls instead of bam files (see User manuals).

For the example data, we also provided the markdown in github.

For Bioconductor, we have submitted the SCATE R package to Bioconductor. It has already passed bioconductor's automatic quality check (see the email attached below) and it is now under the final stage of review by the Bioconductor maintenance team. That review process may take a few weeks to finish. Once approved, the SCATE package will also be available at Bioconductor.

Here is the email that shows our submission to Bioconductor has already passed the initial quality check.

----- Forwarded message -----

From: bioc-issue-bot <notifications@github.com>

Date: Sat, 21 Mar 2020 at 11:31

Subject: Re: [Bioconductor/Contributions] SCATE (#1420)

To: Bioconductor/Contributions <Contributions@noreply.github.com>

Cc: Wenpin Hou <whou10@jhu.edu>, Mention <mention@noreply.github.com>

Dear Package contributor,

This is the automated single package builder at bioconductor.org.

Your package has been built on Linux, Mac, and Windows.

Congratulations! The package built without errors or warnings on all platforms.

Please see the build report for more details.

Second round of review

Reviewer 1

The authors have addressed my previous concerns. It's great that the authors demonstrated SCATE on 10X PBMC data and carried out additional benchmarks.

Reviewer 2

The authors have addressed the mentioned concerns and also improved the usability of the software.